# Intracellular pH dynamics regulates intestinal stem cell lineage specification

Yi Liu [1], Efren Reyes[2,5], David Castillo-Azofeifa[2,3,5], Ophir D. Klein [2], Todd Nystul [4] ✉ & Diane L. Barber [1] ✉

Intracellular pH dynamics is increasingly recognized to regulate myriad cell behaviors. We report a finding that intracellular pH dynamics also regulates adult stem cell lineage specification. We identify an intracellular pH gradient in mouse small intestinal crypts, lowest in crypt stem cells and increasing along the crypt column. Disrupting this gradient by inhibiting $H^+$ efflux by $Na^+/H^+$ exchanger 1 abolishes crypt budding and blocks differentiation of Paneth cells, which are rescued with exogenous WNT. Using single-cell RNA sequencing and lineage tracing we demonstrate that intracellular pH dynamics acts downstream of ATOH1, with increased pH promoting differentiation toward the secretory lineage. Our findings indicate that an increase in pH is required for the lineage specification that contributes to crypt maintenance, establishing a role for intracellular pH dynamics in cell fate decisions within an adult stem cell lineage.

Intracellular pH (pHi) dynamics is increasingly recognized as a key regulator of diverse cell behaviors, including epithelial to mesenchymal transition[1,2], transformation and dysplasia[1,3–5], and differentiation[2,4,6–8]. We previously reported that increased pHi is required for differentiation of clonal mouse embryonic stem cells and *Drosophila* adult ovary follicle stem cells[6,8]. However, it remains unknown whether pHi dynamics regulates lineage specification of mammalian adult stem cells. To investigate this possibility, we used mouse small intestinal organoids as a model. Like the *Drosophila* follicle epithelium, the epithelium of the small intestine is maintained by a pool of adult stem cells (intestinal stem cells, ISCs) that divide regularly during adult homeostasis to self-renew and produce daughter cells that first increase in number through a transit amplification stage and have the potential to differentiate along different lineage trajectories[9–15].

To determine pHi dynamics in ISC behaviors we used 3D organoids derived from the mouse duodenum. Intestinal organoids are a self-sustaining in vitro epithelial 3D model that recapitulates many features of in vivo mammalian intestinal epithelium, including self-renewing stem cells at the base of budding crypts, a distinct crypt-villus architecture, and differentiation and lineage specification of

stem cell progeny[9,14–17]. We identified a pHi gradient along the crypt axis in intestinal organoids as well as in freshly isolated crypts that increases from the base to the top of the crypt column and found that the gradient is maintained by activity of the plasma membrane $H^+$ extruder, $Na^+/H^+$ exchanger 1 (NHE1). Single-cell RNA sequencing (scRNA-seq) revealed that disrupting the pHi gradient impairs specification of the secretory lineage downstream of the master regulator, ATOH1, which we confirmed with lineage tracing. Paneth cells are a major cell type in the secretory lineage, and we found that disrupting the pHi gradient resulted in loss of Paneth cell differentiation, which impairs the WNT circuit and crypt budding. In addition, scRNA-seq revealed a biased stem cell fate decision toward the absorptive lineage and an increased *Clusterin* (*Clu*)$^+$ revival subpopulation. Together, our results reveal a previously unreported role for pHi dynamics as a regulator of mammalian adult stem cell lineage specification.

## Results

### A pHi gradient from NHE1 activity in small intestinal organoids
To quantify the pHi of cells in the ISC lineage, we generated organoids stably expressing a genetically encoded pHi biosensor,

[1]Department of Cell and Tissue Biology, University of California San Francisco, San Francisco, CA 94143, USA. [2]Program in Craniofacial Biology and Department of Orofacial Sciences, University of California San Francisco, San Francisco, CA 94143, USA. [3]Immunology Discovery, Genentech, Inc., South San Francisco, CA 94080, USA. [4]Departments of Anatomy, University of California San Francisco, San Francisco, CA 94143, USA. [5]These authors contributed equally: Efren Reyes, David Castillo-Azofeifa. ✉e-mail: todd.nystul@ucsf.edu; diane.barber@ucsf.edu

mCherry-SEpHluorin[3,6,8,18]. We acquired ratiometric (SEpHluorin/mCherry) images at days 1 and 3 of organoid growth (Fig. 1a, b; Supplementary Fig. 1a) and calibrated fluorescent ratios to pHi values by perfusing with nigericin-containing buffers of known pH values at the end of each imaging set ("Methods"). We found that all cells have a similar pHi at day 1 of organoid growth but by day 3 a pHi gradient develops along the crypt axis (Fig. 1b, c). Calibrating the SEpHluorin/mCherry ratios with a standard curve ("Methods") revealed pHi values at day 3 of ~7.2 in ISCs and Paneth cells, ~7.4 in progenitor cells on the crypt column and ~7.5 in the crypt neck region (Fig. 1c). Ratiometric time-lapse imaging also revealed a similar pHi gradient during crypt formation, with higher pHi in cells within the column and neck than in the crypt base (Fig. 1d; Supplementary Movie 1). To determine whether a similar pHi gradient exists within the intestinal epithelium, we isolated whole crypts from freshly dissected intestines and assayed for pHi using a ratiometric pH-sensitive dye ("Methods"). Indeed, we observed a clear pHi gradient in these freshly isolated crypts, with increasing values extending from the base of the crypt to the top (Fig. 1e, f; Supplementary Fig. 1c). These differences in pHi raised the question of whether pHi dynamics regulates ISC differentiation.

To determine what regulates this pHi gradient, we tested the Na+-H+ exchanger 1 (NHE1), a nearly ubiquitously expressed resident plasma membrane protein that exchanges an influx of extracellular Na+ for an efflux of intracellular H+ and is a key regulator of pHi dynamics in mammalian cells[19–21]. Additionally, the ortholog of NHE1 in *Drosophila*,

*dNhe2*, is necessary for a pHi gradient within the ovarian follicle stem cell lineage, with a lower pHi in follicle stem cells that progressively increases in pre-follicle and mature follicle cells[8]. We found that intestinal organoids treated with 5 μM 5-(N-Ethyl-N-isopropyl)-amiloride (EIPA), a selective pharmacological inhibitor of NHE1[22], had a similar pHi in ISCs as organoids in control conditions at day 1 but significantly lower pHi in ISCs and adjacent non-Paneth cells at day 3. The pHi of EIPA-treated mature Paneth cells was lower at day1 but unaffected at day 3 (Fig. 1b, c). Because the crypts in EIPA-treated organoids did not form buds that were stable enough to have a clear neck region, we could not measure the pHi of crypt neck cells in those organoids. Hence, inhibition of NHE1 activity reduced pHi in most crypt cells and disrupted the pHi gradient along the crypt axis.

## NHE1 activity is necessary for crypt growth

To determine whether the pHi gradient along the crypt axis is functionally significant for organoid development, we inhibited NHE1 activity pharmacologically with EIPA and genetically with a doxycycline (Dox)-inducible CRISPR-Cas9 system ("Methods"). Both of these methods allow acute inhibition of NHE1, which we predict limits compensatory changes that have been reported to arise in animals that are homozygous null for pHi regulators[23–27]. Additionally, although NHE1 has dual functions as a plasma membrane ion transport protein and an actin filament anchor through binding the ERM proteins ezrin. radixin and moesin[28,29], current evidence indicates that

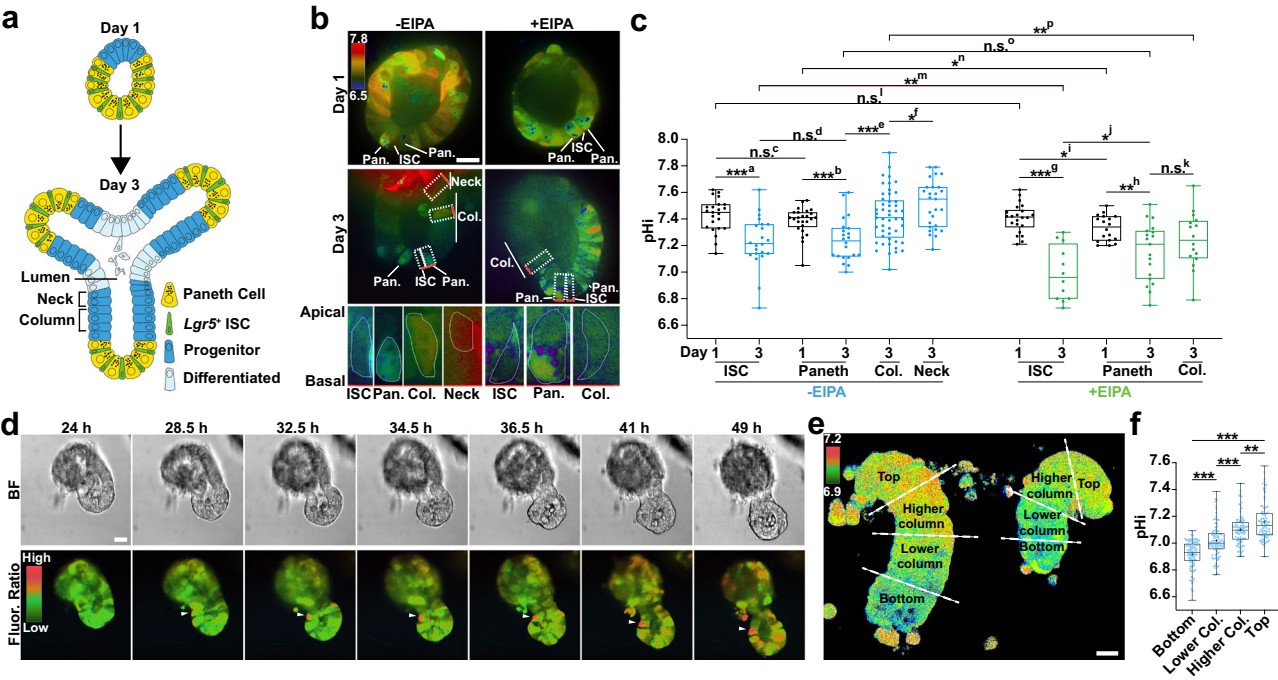

**Fig. 1 | A pHi gradient develops in small intestinal organoid crypts. a** Schematic representation of small intestinal organoids growth. **b–d** pHi in the intestinal crypt of organoids. In (**b**), representative images of mCherry-SEpHluorin ratio in a WT organoid (control) and an WT organoid treated with 5 μM EIPA from day 1 to day 3. Dashed box with red line (basal layer), cells for enlarged view. Bottom panel, enlarged views showing pHi gradient in -EIPA but not in +EIPA. Representative images are from representative 3 preparations. Col., Column. Pan., Paneth cell. In (**c**), values of pHi of different crypt cell types determined by calibrating fluorescent ratios (n = 3 biologically independent experiments; two-sided Mann–Whitney test). Blue, A pHi gradient; Green, attenuated pHi gradient by EIPA. Box plots are minimum to maximum, the box shows 25th-75th percentiles, and the central line is the median. *p* values in -EIPA: [a]*p* = 0.0001; [b]*p* = 0.0001; [c]*p* = 0.2332; [d]*p* = 0.7492; [e]*p* = 0.0002; [f]*p* = 0.0486. *p* values in +EIPA: [g]*p* = 9.222E-06; [h]*p* = 0.0154; [i]*p* = 0.0396; [j]*p* = 0.0468; [k]*p* = 0.2603. *p* values in -EIPAvs+EIPA: [l]*p* = 0.4074;

[m]*p* = 0.0054; [n]*p* = 0.0289; [o]*p* = 0.2609; [p]*p* = 0.0032. Source data are provided as a Source Data file. In (**d**), representative frames (7 of 51) from 15-h time-lapse videos of crypt budding started 24 h after seeding. Arrowheads, growing crypt. BF bright-field view. Fluor. Ratio, ratiometric view of mCherry-SEpHluorin. Time-lapse video see Supplementary Movie 1. **e**, **f** pHi in the isolated crypts. In (**e**), representative ratometric images of freshly isolated crypts loaded with BCECF. Individual fluorescent channels (see Supplementary Fig. 1d). Ratios of BCECF were sampled from bottom, lower column, higher column, and top along the crypt axis. pHi values were calibrated from ratios (n = 2 biologically independent experiments, two-sided Welch's *t* test) and plotted in (**f**). *p* values in (**f**), [BottomvsLower Col.]*p* = 9.753E−09; [Lower Col.vsHigher Col.]*p* = 7.204E−06; [Higher Col.vsTop]*p* = 0.0044; [BottomvsTop]*p* < 2.2E−16. All plots: *p* < 0.05, **p* < 0.01, ***p* < 0.001. n.s., not statistically significant. Source data are provided as a Source Data file. All scale bars represent 20 μm.

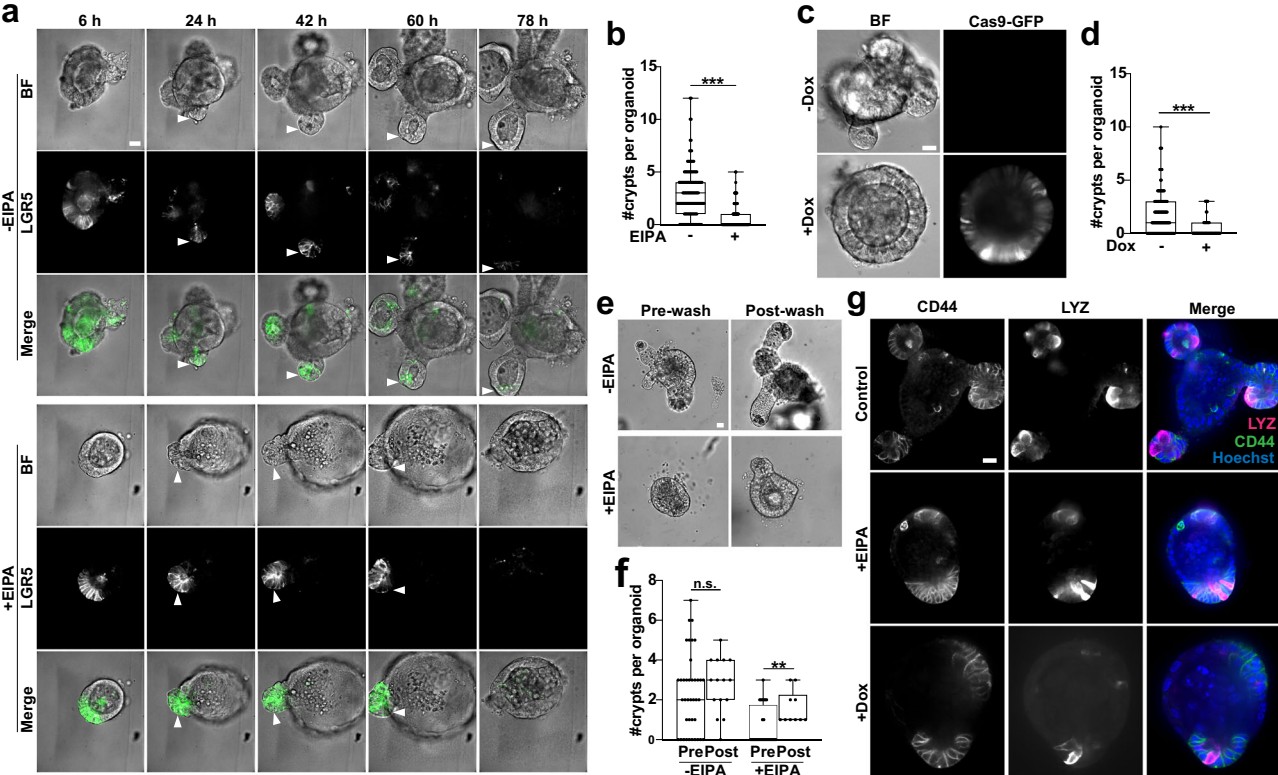

**Fig. 2 | Loss of NHE1 activity attenuates intestinal crypt budding in organoids.**
**a** Representative frames (5 of 25) from 75-h time-lapse microscopy (Supplementary Movie 2, Supplementary Movie 3) showing crypt budding (arrowhead) in $Lgr5^{DTR-GFP}$ organoids from day 0 (seeding) to day 3 in the absence (Control) and presence of 5 mM EIPA ($n = 3$ biologically independent experiments). Control GFP signal is reduced due to photobleaching. BF bright-field view. LGR5, $Lgr5^+$ ISC.
**b** Quantification of crypt number of day 3 organoids treated without (control) and with 5 μM EIPA ($n = 4$ biological independent experiments, two-sided Mann–Whitney test, ***$p < 0.001$, $p = 1.251E-13$). Source data are provided as a Source Data file. **c**, **d** Crypt budding in NHE1-silenced organoids. Representative images of inducible CRISPR-Cas9 mediated NHE1-silenced organoids from $n = 7$ biologically independent experiments (**c**). In (**d**), quantification of crypt number in control and NHE1-silenced organoids ($n = 7$ biologically independent experiments,

two-sided Mann–Whitney test, ***$p < 0.001$, $p = 6.491E-07$). Source data are provided as a Source Data file. **e**, **f** Reversibility of EIPA treatment. In (**e**), representative images of crypt budding in organoids before and after washing. Pre-wash, 1 day in growth medium (without and with EIPA) after seeding. Post-wash, 1 day after washing and reseeding. In (**f**), quantification of crypt number in control and EIPA-treated organoids before and after wash ($n = 3$ biologically independent experiments, two-sided Mann–Whitney test, n.s., not statistically significant: $^{-EIPA\_PrevsPost}p = 0.3031$, **$p < 0.01$: $^{+EIPA\_PrevsPost}p = 0.0017$). Source data are provided as a Source Data file. **g** Representative images of control organoids and organoids lacking NHE1 activity immunolabeled for CD44 to identify crypt cells and lysozyme (LYZ) to identify Paneth cells. All box plots are minimum to maximum, the box shows 25th-75th percentiles, and the median is indicated as a central line. All scale bars represent 20 μm.

EIPA targets only ion transport activity by NHE1 with no effect on actin anchoring.

In control organoids, crypts began budding by day 1, then lengthened and enlarged by day 3 as mature buds (Fig. 2a; Supplementary Movie 2). In contrast, inhibiting NHE1 activity with EIPA or by Dox-inducible CRISPR-Cas9 gene silencing (Fig. 2a, c) significantly reduced the number of budded crypts (Fig. 2b, d). When crypt buds did form in organoids treated with EIPA, they were typically not retained (Fig. 2a; Supplementary Movie 3). Additionally, budding resumed after removing of EIPA at day 3, indicating that organoids remain viable with EIPA treatment (Supplementary Fig. 1b) and that the inhibition of budding is reversible (Fig. 2e, f). To test whether loss of budding is due to the absence of crypt cells, we immunolabeled for the hyaluronic acid receptor CD44, a crypt marker expressed throughout the crypt cell population, including $Lgr5^+$ ISCs, Paneth cells, and crypt progenitors[30–32]. We detected CD44$^+$ cell clusters in organoids that lack NHE1 activity (Fig. 2g), and also identified Paneth cells, as indicated by lysozyme (LYZ)[33] immunolabeling, within the CD44$^+$ population of cells (Fig. 2g), suggesting that these CD44$^+$ clusters are unbudded crypts. Taken together, these data indicate that loss of NHE1 activity disrupts crypt budding but does not eliminate crypt cell populations.

## Single-cell RNA-sequencing of intestinal organoids with loss of NHE1 activity

For a global and unbiased view of NHE1 activity-dependent changes in the intestinal epithelium, we performed scRNA-seq on control and NHE1-inhibited small intestinal organoids using the 10x Genomics platform (Supplementary Fig. 3; "Methods"). After the standard quality control steps ("Methods"; Supplementary Fig. 4a), we merged data from all 4 conditions (WT ± EIPA, CRISPR ± Dox) and performed normalization and batch correction, clustering, cell-type identification, and downstream transcriptional analysis[34]. Each cluster was associated with an enrichment of cell-type specific or cell-cycle signature gene expression, which were then used to specify the identities of each cluster. We observed a strong correlation between our biological replicates (Supplementary Fig. 4b), indicating high reproducibility of our organoid cultures. We visualized the clustering profiles and cell-type gene signatures with UMAP (merged data: Fig. 3a, b; separated data: Supplementary Fig. 4c) and heatmap plots (Fig. 3c) and identified many clusters that were in agreement with previous publications, including generic $Lgr5^+$ ISCs, enterocytes, tuft cells, and enteroendocrine cells (EEC)[6,12,35,36]. In addition, we identified two subsets of $Lgr5^+$ ISCs, a low-proliferative and a high-proliferative subpopulation (Figs. 3b, c, 4a), distinguished by the expression of two reported slowly

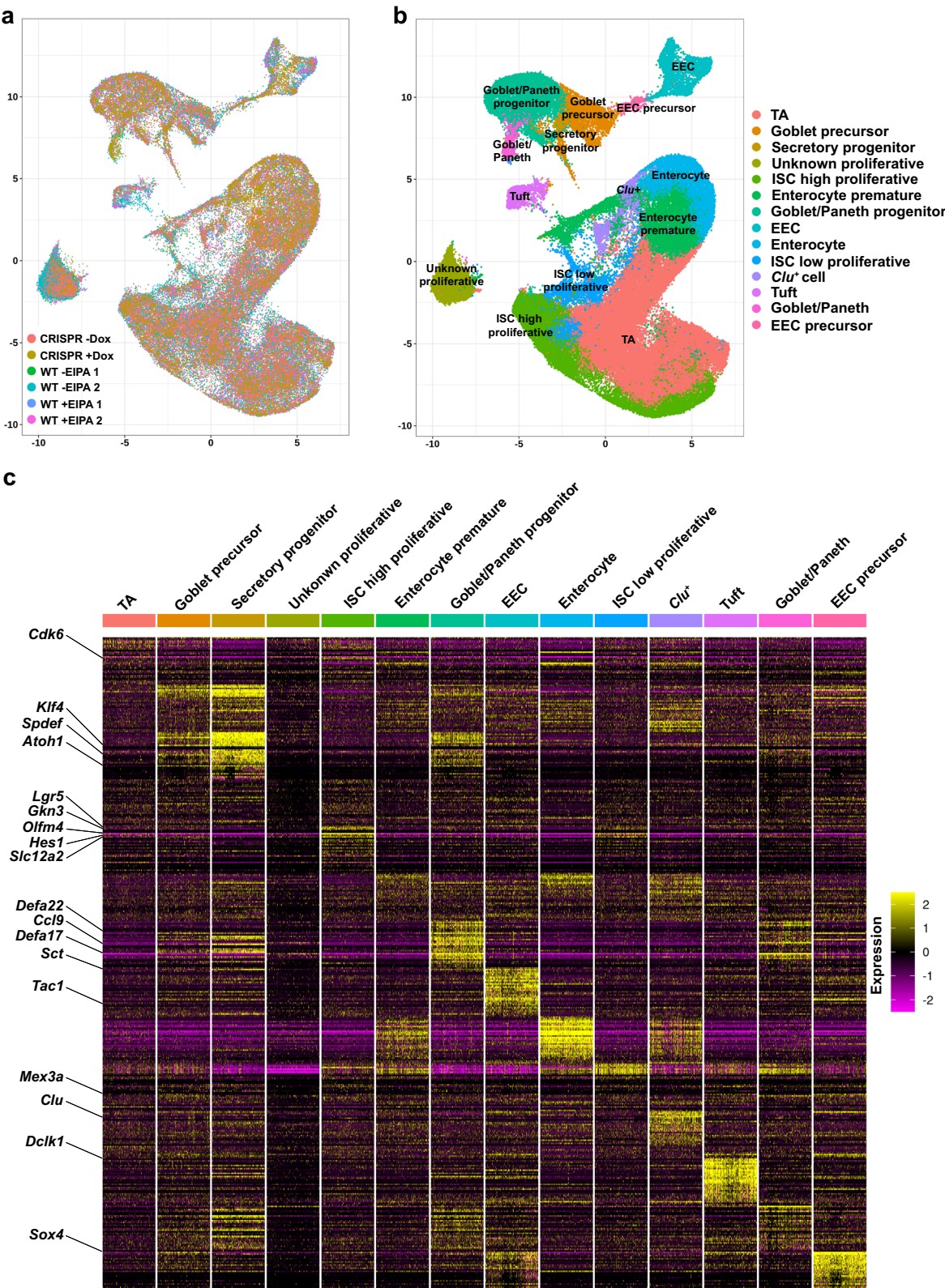

**Fig. 3 | Single-cell transcriptome survey of intestinal organoids. a, b** Uniform manifold approximation and projection (UMAP) visualization of cell profiles from a merged dataset (WT ± EIPA, CRISPR ± Dox). UMAP of joint single-cell profiles (**a**), colored by individual datasets. UMAP of cell-identity clusters based on known markers (**b**), colored by cell identities. **c** Heatmap visualization of expression signatures sorted by cell-identity clusters. The colored expression level of each gene per cell, 0 (population mean) ±2 (Standard Deviation, SD). Conventional cell-type associated genes are highlighted (left row).

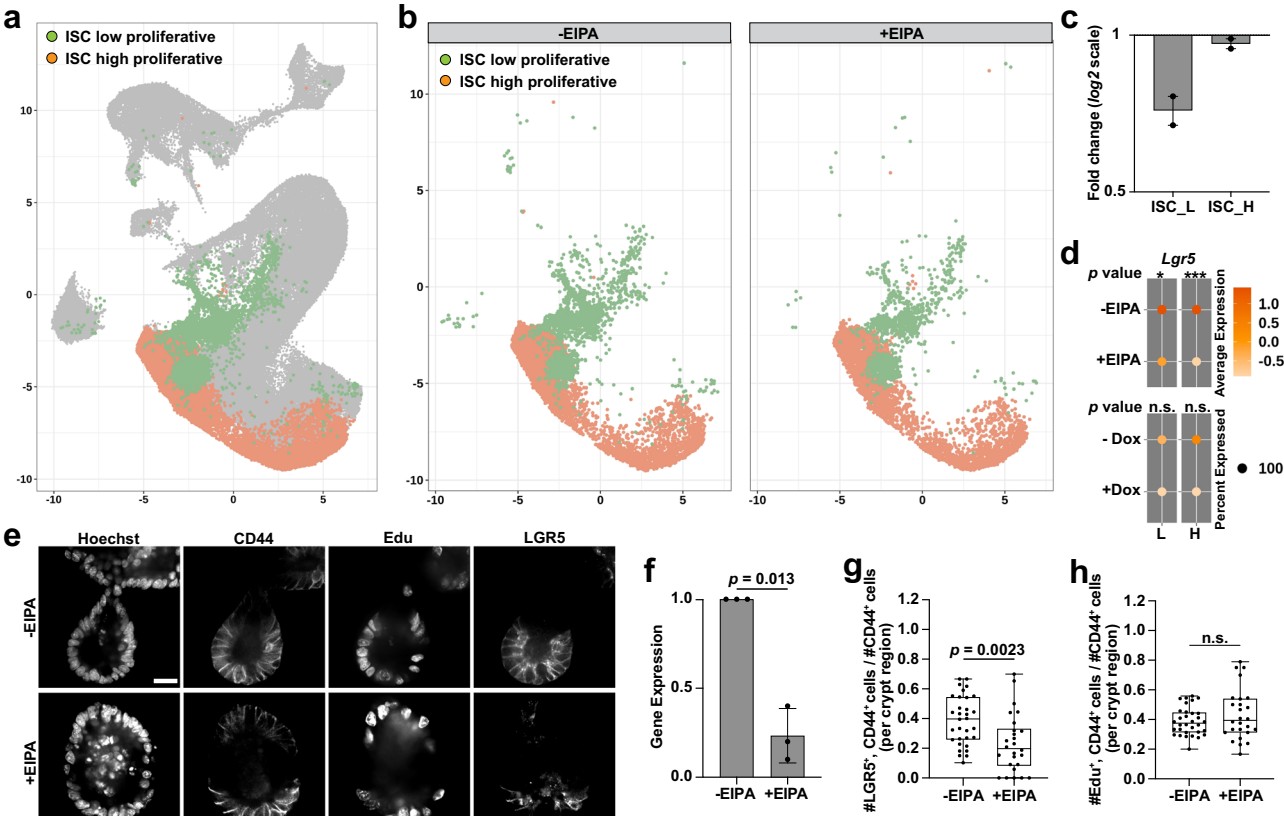

**Fig. 4 | Inhibiting NHE1 activity reduced *Lgr5*⁺ ISCs. a** UMAP plot highlighting distinct *Lgr5*⁺ ISC subtypes. **b** Profiles of *Lgr5*⁺ ISC-subtype clusters without (control) and with EIPA. **c** Fold change of *Lgr5*⁺ ISC subtypes with NHE1 inhibition by EIPA compared to the control (*n* = 2 biologically independent experiments). ISC_L and ISC_H, low proliferative and high proliferative *Lgr5*⁺ ISCs. Scatter dot plot, mean with range. Source data are provided as a Source Data file. **d** Dot plot showing averaged expression level of *Lgr5* in the ISC subtypes. L, low proliferative. H, high proliferative. Statistics, two-sided DECENT test. Colored average expression scale, 0 (population mean) ± SD. \**p* < 0.05, \*\**p* < 0.01, \*\*\**p* < 0.001. n.s., not statistically significant. *p* values: ^L_-EIPAvs+EIPA^*p* = 0.0177; ^L_-Doxvs+Dox^*p* = 0.3708; ^H_-EIPAvs+EIPA^*p* = 3.39E−05; ^H_-Doxvs+Dox^*p* = 0.0874. **e**–**h** Crypt proliferation and *Lgr5*⁺ ISC pool with loss of NHE1 activity. In (**e**), representative images of proliferating (Edu⁺)

and *Lgr5*⁺ ISC (LGR5⁺) populations in the crypt region (CD44⁺) of day 3 organoids (*n* = 5 biologically independent experiments). Scale bar represents 20 μm. In (**f**) Relative *Lgr5* expression (mean ± SD) in control and EIPA-treated organoids via qRT-PCR (*n* = 3 biologically independent experiments; Two-sided Wilcoxon Signed Rank test). Source data are provided as a Source Data file. Quantification of (**g**) *Lgr5*⁺ ISCs (LGR5⁺) and (**h**) number of proliferating cells (Edu⁺) in the crypt region (CD44⁺) (*n* = 5 biologically independent experiments; two-sided student's *t* test; *p* = 0.0524). All box plots are minimum to maximum, the box shows 25th–75th percentiles, and the median is indicated as a central line. All non-dot plots with a statistically significant difference are specified with a *p* value. n.s., not statistically significant. Source data are provided as a Source Data file.

dividing ISC markers, *Mex3a*[37] (Fig. 3c) and *MKi67* (Supplementary Fig. 5a). We also detected differential expression of several other cell cycle genes within these two ISC populations, including *Cdk1*, *Prc1*, and *Cdk6*, and the absorptive lineage-associated gene, *Apoa1* (Supplementary Fig. 5b), consistent with recent evidence of heterogeneously dividing *Lgr5*⁺ ISC pools[37]. In the secretory lineage, we identified distinct clusters of progenitors, an early secretory progenitor, a precursor of both Paneth and Goblet cells, and a Goblet-specific precursor population[10,11,38–41] (Figs. 3b, c, 5a; "Method"). We also found a subcluster associated with the enterocyte lineage markers enriched in *Clusterin* (*Clu*) that is involved in the regeneration response in vivo[17,42,43]. To gain insights into the effects of pHi dynamics, we compared gene expression profiles between the -EIPA and +EIPA conditions and the -Dox and +Dox conditions (Supplementary Fig. 6).

**Inhibiting NHE1 reduced crypt base columnar (CBC) cells**

ISCs in the crypt base are marked by elevated *Lgr5* expression, and datasets from organoids lacking NHE1 activity contained fewer *Lgr5*⁺ ISCs (Fig. 4a–c). In addition, inhibiting NHE1 with EIPA significantly decreased per-cell expression of *Lgr5* within the ISC clusters (Fig. 4d). We also detected a modest but not significant decrease of *Lgr5* in the ISCs of NHE1-silenced organoids (Fig. 4d). This discrepancy is likely due

to the distinct inhibitory mechanisms with EIPA and CRISPR, which are functional complementary but not identical. Consistent with these data, we found decreased *Lgr5* transcript in EIPA-treated organoids compared with control at day 3 by qRT-PCR (Fig. 4f). We also observed a decrease in GFP expression in *Lgr5*^DTR-GFP^ organoids[44] maintained for 3 days with EIPA compared with controls (Fig. 4e; Supplementary Movie 3). To quantify this effect at the cellular level, we immunolabeled for GFP in *Lgr5*^DTR-GFP^ organoids to identify ISCs and CD44 to identify all crypt cells. We found a significant decrease in the percentage of CD44⁺ cells that are *Lgr5*⁺ in organoids maintained with EIPA compared with controls (Fig. 4g). The loss of *Lgr5*⁺ ISCs is known to cause crypt loss in organoids[45], but is not sufficient to impair proliferation in the intestinal epithelium[36,44,46]. Consistent with this, we observed no difference in the frequency of Edu⁺ (Fig. 4e, h) or Ki67⁺ cells (Supplementary Fig. 5c) in organoids maintained without or with EIPA. Together, these findings indicate that inhibiting NHE1 activity causes a decrease in *Lgr5*⁺ ISCs without decreasing proliferation of crypt cells.

**Lowering pHi impairs the secretory fate by acting downstream of ATOH1**

We next determined whether disrupting the NHE1-generated pHi gradient affects the specification of ISC lineages. The ISC lineage

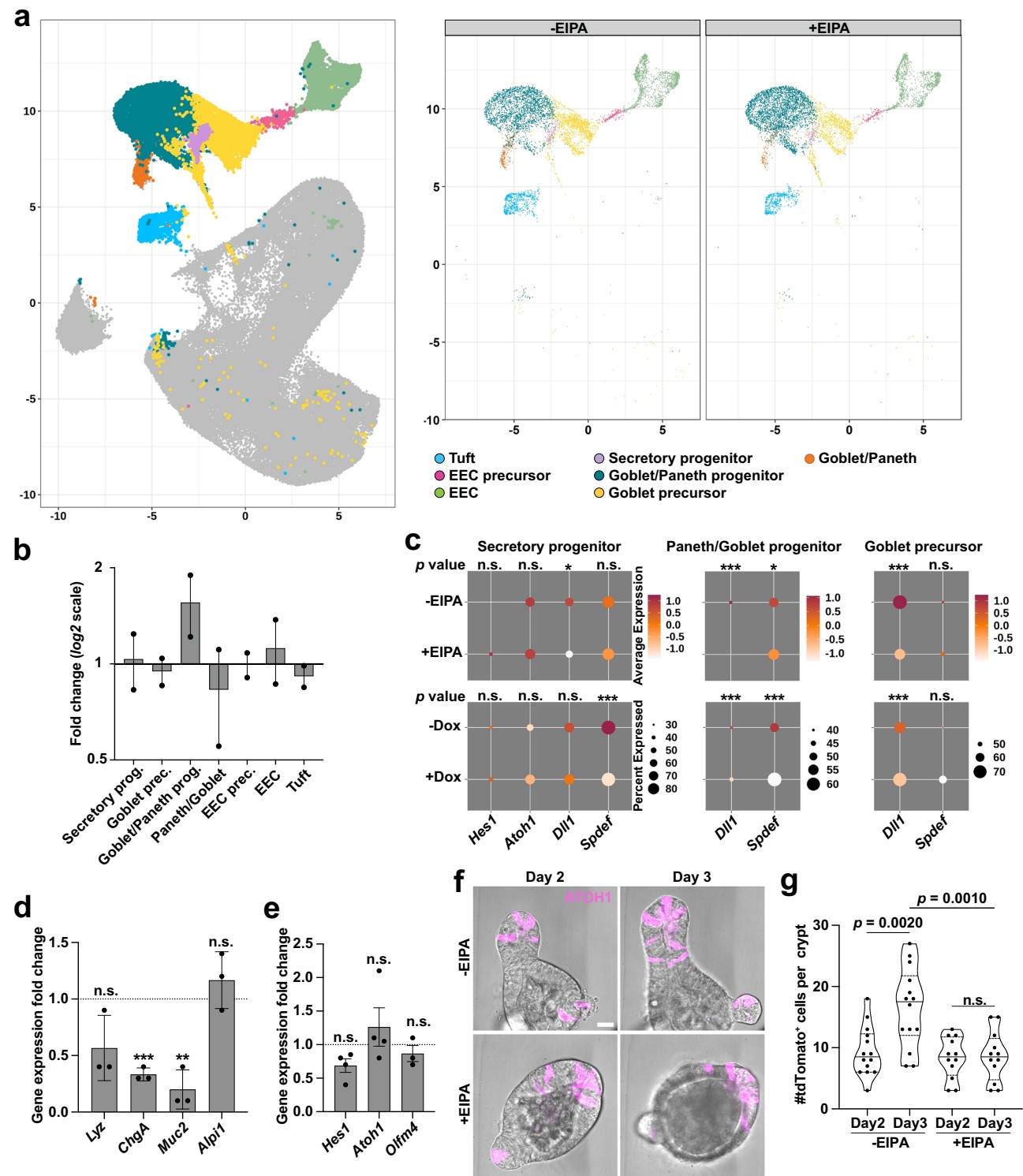

bifurcates into absorptive (enterocyte) and secretory lineages, with the latter including Paneth cells, EEC, and Goblet cells[9,16]. Our analysis of cluster sizes in scRNA-seq datasets revealed an approximately 1.5-fold increase in the percentage of cells classified as Paneth-Goblet progenitors and a 1.2-fold decrease in the percentage of cells in the Differentiated Paneth/Goblet cell cluster in the EIPA-treated organoids compared with controls (Fig. 5a, b). Specification of the secretory lineage is regulated by Notch and ATOH1 signaling pathway activity in the daughter cells of ISCs[12,40,47,48]. Notch lateral inhibition causes immediate daughter cells from ISCs to adopt a state of either high or low Notch pathway activity[12,48]. High levels of Notch signaling promote

the expression of *Hes1*, which represses the master secretory lineage transcription factor ATOH1 and thus promotes differentiation toward the absorptive cell fate. Conversely, in cells with low Notch pathway activity, low levels of HES1 permits *Atoh1* expression, which promotes the secretory cell fate. ATOH1 forms a positive feedback loop by reinforcing its own expression and promoting the expression of Delta-like ligands, including *Dll1*, thus enabling lateral inhibition of the secretory fate in neighboring cells[12,40,47–50]. We found that expression of *Hes1* and *Atoh1* were similar in secretory progenitor cell populations from control and experimental datasets (Fig. 5c), suggesting that inhibiting NHE1 activity does not impair Notch pathway activity in

**Fig. 5 | Loss of NHE1 activity impaired the secretory lineage. a** UMAP plot highlighting secretory clusters. left, merged view; right, splited view of -EIPA (Control) and +EIPA. **b** Fold change of secretory cell types with EIPA-mediated NHE1 inhibition compared to the control ($n = 2$ biologically independent experiments). Scatter dot plot, mean with range. Source data are provided as a Source Data file. **c** Dot plots showing averaged expression level of secretory associated genes in the progenitors and precursor. Two-sided DECENT test. $p$ for secretory progenitor: $^{Hes1}$-EIPAvs+EIPA$p = 0.2056$; $^{Hes1}$-Doxvs+Dox$p = 0.9409$; $^{Atoh1}$-EIPAvs+EIPA$p = 0.6288$; $^{Atoh1}$-Doxvs+Dox$p = 0.8094$; $^{Dll1}$-EIPAvs+EIPA$p = 0.01419$; $^{Dll1}$-Doxvs+Dox$p = 0.4880$; $^{Spdef}$-EIPAvs+EIPA$p = 0.4191$; $^{Spdef}$-Doxvs+Dox$p = 0.0002126$. $p$ for Paneth/Goblet progenitor: $^{Dll1}$-EIPAvs+EIPA$p = 3.02E-13$; $^{Dll1}$-Doxvs+Dox$p = 1.06E-07$; $^{Spdef}$-EIPAvs+EIPA$p = 0.01171$; $^{Spdef}$-Doxvs+Dox$p = 0.0006446$. $p$ for Goblet precursor: $^{Dll1}$-EIPAvs+EIPA$p = 1.44E-13$; $^{Dll1}$-Doxvs+Dox$p = 0.0001124$; $^{Spdef}$-EIPAvs+EIPA$p = 0.79608912$; $^{Spdef}$-Doxvs+Dox$p = 0.3181$. **d** Cell marker expression (mean ± SD) in EIPA-treated organoids relative to controls by qRT-PCR ($n = 3$ biologically independent experiments, two-sided paired $t$ test). *Lyz*, lysozyme, *ChgA*, chromogranin A, *Muc2*, mucin 2, *Alpi1*, alkaline phosphatase. $p$ values: $^{Lyz}p = 0.1215$; $^{ChgA}p = 0.0025$; $^{Muc2}p = 0.0153$; $^{Alpi1}p = 0.3701$. Source data are provided as a Source Data file. **e**, Relative gene expression (mean ± SD) of Notch signaling pathway without and with EIPA by qRT-PCR (*Hes1*, *Atoh1*: $n = 4$ biologically independent experiments; *Olfm4*: $n = 3$ biologically independent experiments, two-sided One sample $t$ test). $p$ values: $^{Hes1}p = 0.0533$; $^{Atoh1}p = 0.4275$; $^{olfm4}p = 0.3828$. Source data are provided as a Source Data file. **f, g** Lineage tracing for secretory cells in *Atoh1*$^{CreERT2}$;*Rosa26*$^{tdTomato}$ organoids without and with EIPA. Representative frames (day 2 and day 3) from a time-lapse recording show secretory cells indicated by ATOH1$^+$ (tdTomato$^+$) in organoids (**f**). Scale bar, 20 μm. Movie see Supplementary Movie 4 and Supplementary Movie 5. In (**g**), quantification of ATOH1$^+$ (tdTomato$^+$) labeled secretory cells from day 2 to day 3 within a crypt region ($n = 3$ biologically independent experiments, two-sided Welch's $t$ test). $p$ with statistically significant difference is specified in (**g**). $^{+EIPA\_Day\ 2vsDay\ 3}p = 0.8719$. All plots: *$p < 0.05$, **$p < 0.01$, ***$p < 0.001$. n.s., not statistically significant. Source data are provided as a Source Data file.

these cells. In contrast, expression of one or more ATOH1 key targets that are known to be involved in the secretory fate specifications, including *Dll1*, *Spdef*, *Gfi1*, and *Neurog3*[38,39,41,51,52] was decreased in the secretory progenitor, Paneth/Goblet progenitor, and Goblet precursor cell populations from EIPA-treated organoids (Fig. 5c, Supplementary Fig. 8a). We did not, however, observe a significant decrease in most of the non-secretory fate responsive targets of ATOH1 (Supplementary Fig. 8a), suggesting the pH sensing acting downstream of *Atoh1* expression is selective and specific for the secretory lineage. We also found that several genes typically associated with the absorptive fate were elevated in multiple secretory cell types with loss of NHE1 activity, including secretory progenitors and the Goblet cell precursors (Fig. 6d), suggesting that NHE1 activity might reinforce the distinction between the absorptive and secretory lineages.

To further test the prediction that lowering pHi impairs downstream functions of ATOH1 and secretory lineage specification, we profiled global changes of secretory markers by qRT-PCR and found that expression of two of the three secretory cell markers—*Chromogranin A* (*ChgA*) for EEC cells, and *Muc2* for Goblet cells—was significantly decreased in EIPA-treated organoids compared with controls (Fig. 5d). Consistent with reduced transcripts, inhibiting NHE1 activity decreased the number of EECs in organoids as determined by immunolabeling for Chromogranin A (CHGA) (Supplementary Fig. 8b). In addition, we found no statistically significant change in the expression of *Atoh1* and Notch responsive genes *Hes1* and *Olfm4* in EIPA-treated organoids compared with controls as determined by qRT-PCR (Fig. 5e), consistent with our observations of the scRNA-seq data (Fig. 5c). These data suggest that loss of NHE1 activity impairs secretory lineage specification by affecting differentiation downstream of *Atoh1* expression. To test this hypothesis, we performed lineage tracing with *Atoh1*$^{CreERT2}$; *Rosa26*$^{tdTomato}$ organoids[10] (Fig. 5f; Supplementary Movie 4, Supplementary Movie 5). We found no difference in the number of tdTomato$^+$ cells between control and EIPA-treated organoids on Day 2 (Fig. 5g), indicating the rates of clone induction in ATOH1$^+$ cells were similar in both conditions. However, although there was a net increase of tdTomato$^+$ cells over time in control organoids, as the number of tdTomato$^+$ cells per crypt remained constant in organoids treated with EIPA (Fig. 5g; Supplementary Fig. 7c; Supplementary Movie 4, Supplementary Movie 5), suggesting that differentiation and clone expansion from an *Atoh1*$^+$ progenitor was attenuated. As an additional test of the effect of inhibiting NHE1 on secretory cell fate specification, we performed lineage-tracing with *Lgr5*$^{CreER}$;*Rosa26*$^{RFP}$ organoids[10] and assayed for RFP$^+$ cells that had differentiated into DLL1$^+$ secretory progenitors (Fig. 6a). We found that the frequency (Fig. 6b) and number (Fig. 6c) of RFP$^+$, DLL1$^+$ cells in the CD44$^+$ region were significantly less in EIPA-treated organoids compared with controls. Taken together, these data indicate that loss of NHE1 activity impairs the production of mature secretory cells downstream of ATOH1.

## Loss of NHE1 activity and decreasing pHi retains absorptive lineage specification

We next tested whether the loss of NHE1 activity affects specification of the absorptive lineage. We found that a large number of cells were classified as enterocytes in both the -EIPA and +EIPA datasets (Fig. 7a), and observed that the enterocyte cluster constituted an approximately 2-fold higher percentage of the total cell population on average in the +EIPA datasets compared with the control (-EIPA) datasets (Fig. 7b, c). In addition, we observed a significant increase in the mean expression levels of *Alpi1*, *Aldolase B* (*Aldob*), and *Apoa1* within the enterocyte clusters of the +EIPA datasets compared with the control datasets (Fig. 7d). Similarly, we detected a significant increase of *Alpi1*, *Aldob*, *Apoa1*, and *Apoa4* transcripts in the enterocyte clusters with genetic loss of NHE1 (Fig. 7d). However, we did not detect a significant difference in *Alpi1* transcript levels by qRT-PCR using extracts from entire organoids maintained without or with EIPA (Fig. 5d). As an additional test for an effect of EIPA on specification of the absorptive cell specification, we performed lineage tracing with *Lgr5*$^{CreER}$;*Rosa26*$^{RFP}$ organoids, which traces the differentiation of progeny from *Lgr5*$^+$ ISCs into all cell types of the lineage (Supplementary Fig. 7a), including enterocytes (Fig. 7g), secretory progenitors (Fig. 6a), and Paneth cells (Fig. 8a). We observed no difference in the number of enterocytes, marked by *Aldob* expression[17,53] within RFP-labeled clones in EIPA-treated organoids compared with controls (Fig. 7g–i; Supplementary Fig. 7d). Taken together, these results indicate that inhibiting NHE1 activity does not impair absorptive cell differentiation and might induce a slight bias in differentiation toward this lineage. Notably, we also observed a 4-fold increase in the percentage of cells classified as *Clu*$^+$ within a subset of enterocyte clusters from EIPA-treated organoids (Fig. 7b, c) and a significant increase in *Clu* expression in organoids without NHE1 activity (Fig. 7e, f). Emergence of *Clu*$^+$ cells is a signature of intestine regeneration in vivo[17,42,43], raising the possibility that inhibiting of NHE1 activity may trigger a regeneration-like response in organoids.

## Loss of NHE1 activity impairs Paneth cell differentiation

Paneth cells are a major cell type in the secretory lineage and are required for ISC self-renewal in organoid cultures. We therefore tested the prediction that loss of NHE1 activity impairs Paneth cell differentiation by generating clones in *Lgr5*$^{CreER}$;*Rosa26*$^{RFP}$ organoids and immunolabeling for the Paneth cell marker, LYZ[33]. We found that loss of NHE1 activity significantly decreased the number of LYZ$^+$ Paneth cells in the RFP$^+$ clones generated from *Lgr5*$^+$ ISCs (Fig. 8a–c). Even in EIPA-treated organoids in which nearly all the cells were RFP$^+$, Paneth cells remained negative for RFP (Supplementary Fig. 7b). These findings indicate that Paneth cell production by *Lgr5*$^+$ ISC is strongly impaired with loss of NHE1 activity. To further test the effect of

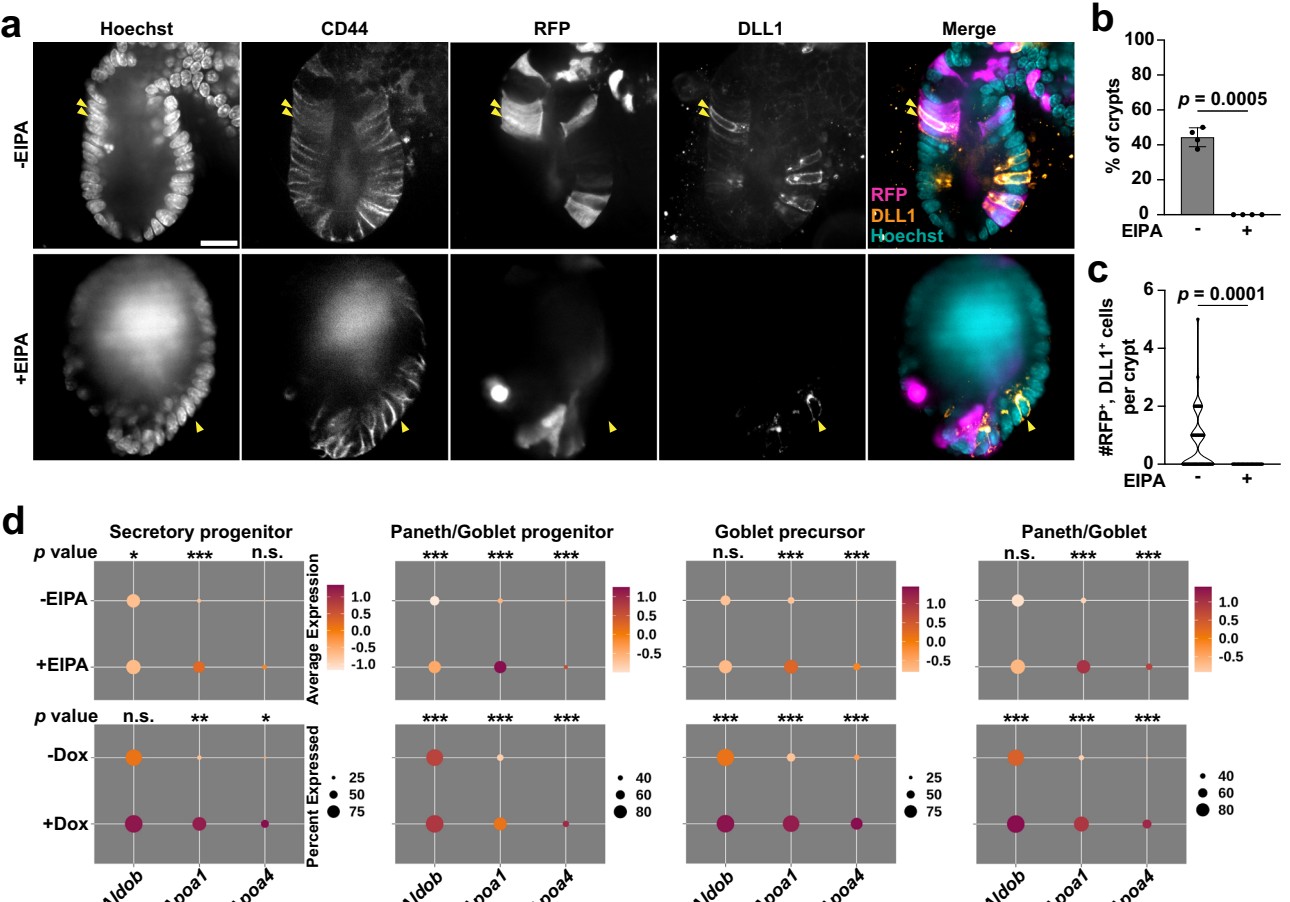

**Fig. 6 | Loss of NHE1 activity impaired the secretory progenitor fate specification. a–c** Lineage tracing for secretory progenitors from *Lgr5*[+] ISCs using *Lgr5*[CreER];*Rosa26*[RFP] organoids. In (**a**), representative images from 4 preparations, show double labeling of *Lgr5*[+] ISC progeny expressing RFP and immunolabeled for DLL1 (arrowhead) in the crypt region, indicated by CD44[+] immunolabeling, of day 2 control but not EIPA-treated organoids. Scale bar, 20 μm. In (**b**), the percent (mean ± SD) of *Lgr5*[+] ISC labeled crypt region at day 2 organoids containing RFP[+]/DLL1[+] cells in the absence or presence of EIPA (*n* = 4 biologically independent experiments, two-sided Mann–Whitney test). Source data are provided as a Source Data file. In (**c**), quantification of the number of RFP[+]/DLL1[+] cells in the crypt region in day 2 organoids maintained in the absence or presence of EIPA (*n* = 4 biologically independent experiments, two-sided Mann–Whitney test). Source data are provided as a Source Data file. Data in (**b**) and (**c**) with a statistically significant difference are specified with a *p* value. **d** Dot plots showing the average expression level of

absorptive lineage markers in the secretory cells. All dot plots are colored by average expression, 0 (population mean) ± SD. Data are analyzed by two-sided DECENT test. All dot plots: \**p* < 0.05, \*\**p* < 0.01, \*\*\**p* < 0.001. n.s., not statistically significant. *p* for secretory progenitor: [*Aldob*]-EIPAvs+EIPA*p* = 0.01236; [*Aldob*]-Doxvs+Dox*p* = 2.04E−05; [*Apoa1*]-EIPAvs+EIPA*p* = 5.76E−05; [*Apoa1*]-Doxvs+Dox*p* = 4.72E−68; [*Apoa4*]-EIPAvs+EIPA*p* = 0.1824; [*Apoa4*]-Doxvs+Dox*p* = 1.02E−29. *p* for Paneth/Goblet *p*rogenitor: [*Aldob*]-EIPAvs+EIPA*p* = 4.65E−17; [*Aldob*]-Doxvs+Dox*p* = 2.04E−05; [*Apoa1*]-EIPAvs+EIPA*p* = 5.99E−116; [*Apoa1*]-Doxvs+Dox*p* = 4.72E−68; [*Apoa4*]-EIPAvs+EIPA*p* = 3.34E−55; [*Apoa4*]-Doxvs+Dox*p* = 1.02E−29. *p* for Goblet precursor: [*Aldob*]-EIPAvs+EIPA*p* = 0.4738; [*Aldob*]-Doxvs+Dox*p* = 3.95E−07; [*Apoa1*]-EIPAvs+EIPA*p* = 8.85E−89; [*Apoa1*]-Doxvs+Dox*p* = 1.43E−53; [*Apoa4*]-EIPAvs+EIPA*p* = 6.32E−17; [*Apoa4*]-Doxvs+Dox*p* = 1.66E−19. *p* for Paneth/Goblet: [*Aldob*]-EIPAvs+EIPA*p* = 0.4077; [*Aldob*]-Doxvs+Dox*p* = 1.31E−06; [*Apoa1*]-EIPAvs+EIPA*p* = 3.59E−14; [*Apoa1*]-Doxvs+Dox*p* = 1.32E−25; [*Apoa4*]-EIPAvs+EIPA*p* = 0.0002154; [*Apoa4*]-Doxvs+Dox*p* = 2.28E−07. All violin plots are minimum to maximum, the dashed line shows 25th-75th percentiles, and the median is indicated as a central line.

inhibiting NHE1 activity on the production of Paneth cells from ATOH1[+] secretory progenitors, we generated clones in *Atoh1*[CreERT2]; *Rosa26*[tdTomato] organoids and traced the production of Paneth cells over the course of 2 days in timelapse movies. We identified cells in the clones based on endogenous tdTomato[+] fluorescence and Paneth cells based on their dense cytoplasmic granules, which are visible with bright-field microscopy. To capture the rate of new Paneth cell production, we assayed for differences between Paneth cell number from day 1 to day 2 (Fig. 8d–f). In control organoids, the number of Paneth cells significantly increased, as expected, whereas with EIPA the number of tdTomato[+] Paneth cells remained constant (Fig. 8e, f). Hence, inhibiting NHE1 activity impaired Paneth cell differentiation from an ATOH1[+] secretory progenitor.

As an additional test for effects on Paneth cells, we immunolabeled for CD44 and the Paneth cell marker, LYZ (Fig. 8g) in organoids maintained without or with EIPA. The number of LYZ[+] cells significantly

increased from day 1 to day 3 in control organoids but remained unchanged over this period with EIPA (Fig. 8h). These data indicate that new Paneth cells accumulate in the growing crypt under normal conditions, consistent with the idea that constant replenishment is important to maintain the Paneth cell population in the crypt, and that NHE1 activity is required for this process. Taken together, our findings show that loss of NHE1 activity impairs Paneth cell fate specification, therefore decreasing the Paneth cell replenishment and reducing Paneth cell numbers in the growing crypt.

**Exogenous WNT rescues crypt budding impaired with loss of NHE1 activity**

In organoids maintained in standard growth medium without WNT, Paneth cells are the only source of WNT ligands for ISCs[15,33]. To test whether loss of NHE1 activity impaired WNT signaling in organoids, we first immunolabeled for EPHB2, which is a WNT pathway target and

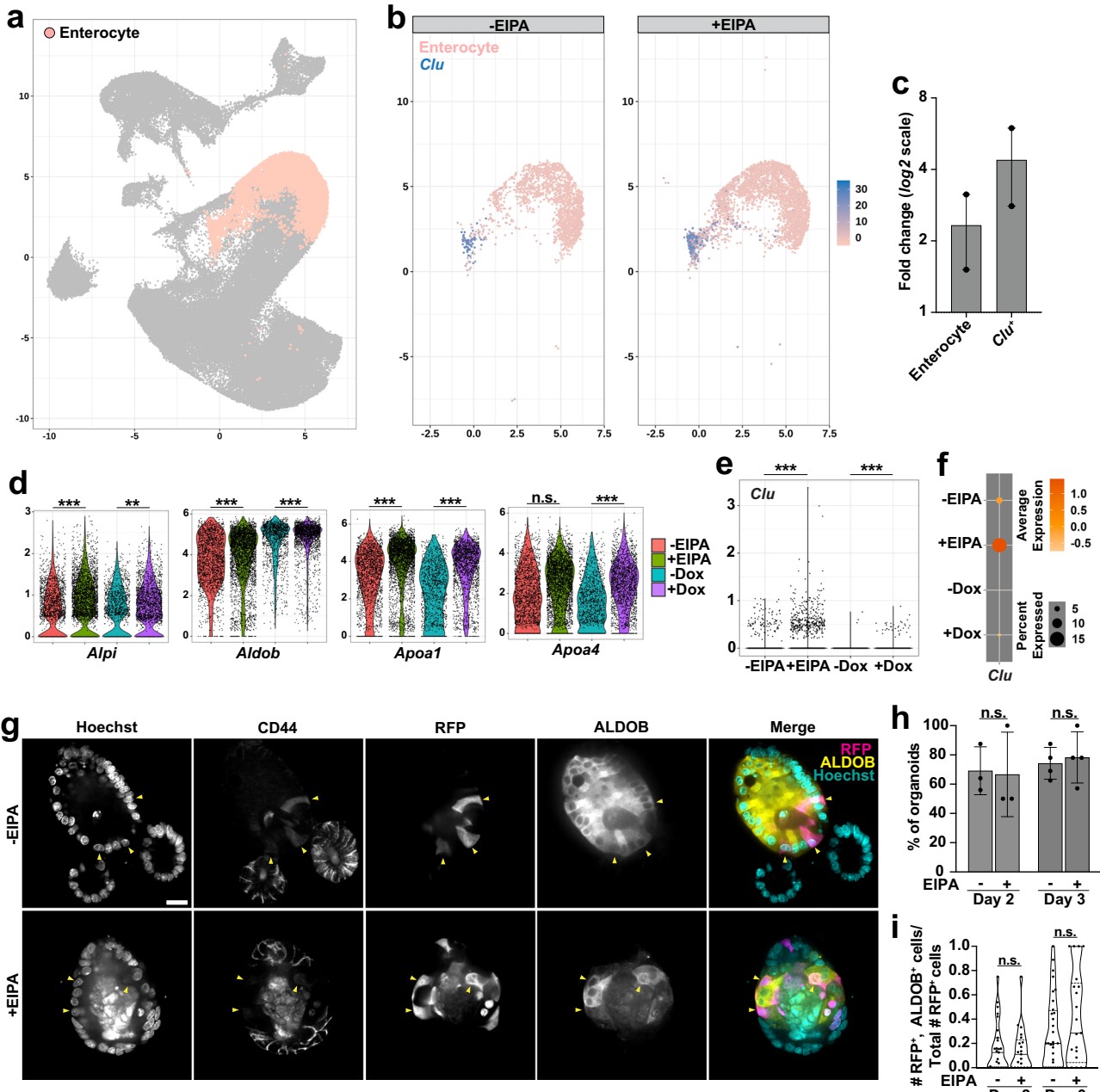

**Fig. 7 | Loss of NHE1 activity retains the absorptive lineage specification.**
**a** UMAP plot highlighting the enterocyte cluster. **b** Profiles of the enterocyte clusters without and with EIPA. The expression of *Clu* is detected in a subset of enterocyte clusters. **c** Fold change of enterocytes and *Clu*+ cells with NHE1 inhibition compared to the control (*n* = 2 biologically independent experiments). Scatter dot plot, mean with range. Source data are provided as a Source Data file. Violin plots showing expression levels of enterocyte markers (**d**) or *Clu* (**e**) in the enterocyte clusters with or without NHE1 inhibition. Y-axis, normalized RNA counts; Two-sided DECENT test. *$p$ < 0.05, **$p$ < 0.01, ***$p$ < 0.001. n.s., not statistically significant. $p$ values in (**d**): *Alpi*_-EIPAvs+EIPA$p$ = 3.22E−08; *Alpi*_-Doxvs+Dox$p$ = 0.002140; *Aldob*_-EIPAvs+EIPA$p$ = 5.77E−07; *Aldob*_-Doxvs+Dox$p$ = 2.18E−05; *Apoa1*_-EIPAvs+EIPA$p$ = 2.11E−16; *Apoa1*_-Doxvs+Dox$p$ = 1.19E−07; *Apoa4*_-EIPAvs+EIPA$p$ = 0.4272; *Apoa4*_-Doxvs+Dox$p$ = 1.19E−10. $p$ values in (**e**): *Clu*_-EIPAvs+EIPA$p$ = 5.21E−13; *Clu*_-Doxvs+Dox$p$ = 8.95E−13. **f** Dot plot showing the *Clu* expression in the enterocyte clusters without and with NHE1 inhibition. Colored average expression, 0 (population mean) ± SD. **g–i** Lineage tracing for enterocytes

in *Lgr5*^CreER;*Rosa26*^RFP organoids (Day 3) without and with EIPA. In (**g**), representative images from 4 preparations show labeling of *Lgr5*+ ISC progeny expressing RFP and ALDOB (arrowhead) in the villus region, indicated by negative CD44 immunolabeling. Scale bars, 20 μm. In (**h**), the percent of organoids (mean ± SD) at which the villus region contains RFP+ and ALDOB+ cells (day 2: *n* = 3 biologically independent experiments, day 3: *n* = 4 biologically independent experiments, two-sided Mann–Whitney test). ^Day 2^$p$ = 0.6000; ^Day 3^$p$ = 0.8857. Source data are provided as a Source Data file. In (**i**), Quantification of the number of RFP+, ALDOB+ double-labeled cells in the villus region of day 2 and day 3 organoids (day 2: *n* = 3 biologically independent experiments, day 3: *n* = 4 biologically independent experiments, two-sided Mann–Whitney test). ^Day 2^$p$ = 0.8791; ^Day 3^$p$ = 0.3310. Violin plots in (**i**) are minimum to maximum, the dashed line shows 25th–75th percentiles, and the median is indicated as a central line. n.s., not statistically significant. Source data are provided as a Source Data file.

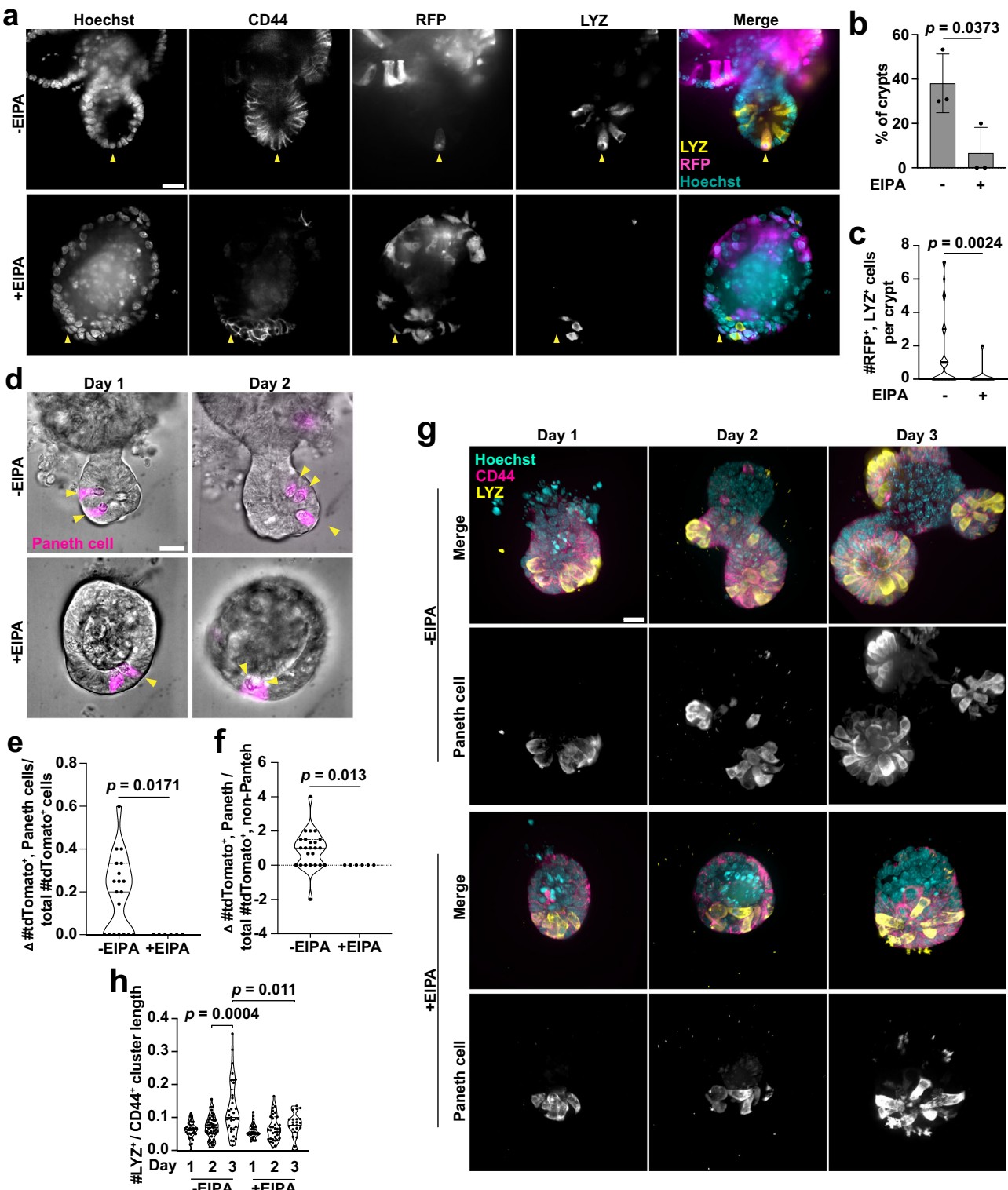

expressed throughout the crypt[54,55]. We found markedly less EPHB2 labeling in EIPA-treated organoids compared with controls (Fig. 9a; Supplementary Fig. 10b). In addition, we examined the expression levels of a broad range of WNT pathway targets (*Axin2, Acl2, Klf5, Smoc2, Jun, Ccnd1, Cdx1, Myc, Rnf43, Tfap4, Vegfa, Tbx3, Efnb1-3*) in the ISC clusters of the scRNA-seq datasets. Although we found two canonical WNT genes, *Axin2* and *Ascl2*, were not decreased with loss of NHE1 activity as confirmed by qRT-PCR (Supplementary Fig. 11a, b), most genes in this panel were downregulated in organoids lacking NHE1 activity (Supplementary Fig. 11c). Previous studies identified

multiple points in the WNT pathway that are regulated, either positively or negatively, by changes in pHi[56–58]. Given this complex relationship, it is likely that the effects of pHi on WNT signaling are context specific, which likely accounts for not all genes in this panel being equally affected. Nonetheless, the data support a model in which inhibiting NHE1 activity generally impairs WNT signaling in ISCs.

Although Paneth cell number is reduced in EIPA-treated organoids, they are not completely absent. Therefore, to test whether decreased WNT signaling is due to an inability of the existing Paneth cells to induce WNT signaling in ISCs, we performed a single cell-based

**Fig. 8 | Loss of NHE1 activity impaired Paneth cell differentiation. a–c** Lineage tracing for Paneth cells from *Lgr5*⁺ ISCs without and with NHE1 inhibition. In (**a**), representative images showing the presence *Lgr5*⁺ ISC progeny expressing RFP and lysozyme (LYZ) (arrowhead) in the crypt region (CD44⁺) of day 3 control but not the EIPA-treated organoids. In (**b**), percent of crypt region (mean ± SD) in day 3 organoids contains RFP⁺, LYZ⁺ cells (*n* = 3 biologically independent experiments, two-sided Mann–Whitney test). Source data are provided as a Source Data file. In (**c**), quantification of the number of RFP⁺, LYZ⁺ cells in the crypt region of day 3 organoids (*n* = 3 biologically independent experiments, two-sided Mann–Whitney test). Source data are provided as a Source Data file. **d–f** lineage tracing for newly produced Paneth cells from the secretory progenitors during the development of *Atoh1*^CreERT2^;*Rosa26*^tdTomato^ organoids in the absence and presence of EIPA. In (**d**), representative frames (day 1 and day 2) from the time-lapse recordings. Newly produced Paneth cells are indicated by the presence of visible dense granules (arrowhead) that are also marked by the secretory lineage marker ATOH1

(tdTomato⁺). Data from *n* = 3 biologically independent experiments are normalized to either (**e**) the number of all labeled secretory cells (tdTomato⁺), or (**f**) the number of non-Paneth secretory cells (tdTomato⁺, non-Paneth) on day 1. Two-sided Mann–Whitney test. Source data are provided as a Source Data file. **g, h** Paneth cell abundance in crypts during organoid development from day 1 to day 3. In (**g**), representative 3D confocal images show Paneth cells in organoid crypts at the indicated days after plating without and with EIPA. CD44, crypt region. LYZ, Paneth cell. Quantification of Paneth cell numbers (**h**) from conditions described (**g**) (*n* = 3 biologically independent experiments; two-sided Mann–Whitney test and two-sided student's *t* test). Source data are provided as a Source Data file. All violin plots are minimum to maximum, the box shows 25th-75th percentiles, and the median is indicated as a central line. Data in plots with a statistically significant difference specified with *p* values, otherwise are not significantly different. All scale bars, 20 μm.

in vitro reconstitution assay[33,59] (Supplementary Fig. 9). We found that mixing single WT *Lgr5*⁺ ISCs with single NHE1-silenced Paneth cells in the absence of exogenous WNT3A did not reduce the efficiency of organoid formation (Supplementary Fig. 10a). These data suggest that the reduction in WNT signaling is due to the decreased Paneth cell number rather than an impaired ability of mature Paneth cells to secrete WNT ligands. Taken together with our finding that inhibiting NHE1 does not impair the ability of Paneth cells to promote ISC self-renewal (Supplementary Fig. 10a), these results suggest that the decrease in ISC number and impaired crypt budding in the absence of NHE1 activity is likely a direct consequence of the loss of Paneth cell differentiation.

As Paneth cells provide essential WNT ligands to ISCs, we next tested whether restoring WNT signaling is sufficient to rescue the ISC maintenance and crypt budding phenotypes with inhibiting NHE1 activity. Adding either 20% WNT3A ligand or 3X RSPO1 to the medium increased the ISC number and crypt budding in the presence of EIPA (Fig. 9a–e), indicating a rescued phenotype. Notably, 20% WNT3A is a relatively low concentration that is not sufficient to induce growth of spheroid-shaped organoids. Additionally, we observed that production of the WNT-responsive gene EPHB2 was restored to control levels, which confirms that WNT signaling is abrogated by inhibiting NHE1 activity (Fig. 9a; Supplementary Fig. 10b). Taken together, our findings support a model in which an NHE1 activity-dependent pHi gradient that acts downstream of ATOH1 function in the progenitors facilitates crypt homeostasis by regulating secretory lineage specification (Fig. 9f).

## Discussion

We identified a pHi gradient in mouse small intestinal organoids and freshly isolated mouse small intestine, with a lower pHi in cells at the crypt base and a progressive increase along the crypt column. Attenuating this gradient in organoids by pharmacologically or genetically inhibiting activity of NHE1, a plasma membrane H⁺ extruder, inhibited the cell-fate decision of *Lgr5*⁺ ISCs toward the secretory lineage. This altered lineage specification occurs in part by reducing ATOH1-dependent signaling and thus decreasing Paneth cell differentiation, which is essential for crypt budding. Moreover, these findings are supported by scRNA-seq and confirmed experimentally by qRT-PCR, immunolabeling, single cell reconstitution, and lineage tracing.

Increasing evidence indicates a previously unrecognized role for pHi dynamics in the differentiation of diverse types of stem cells. Increased pHi is necessary for the differentiation of clonal mouse embryonic stem cells from naïve to primed states[8], mesenchymal stems cells to cardiomyocytes[60], and CD4⁺ T helper 9 cells[61]. Additionally, in vivo studies confirm that increased pHi is necessary for the differentiation of *Drosophila* follicle stem cells[6,8], melanocytes during zebrafish neural crest development[2], and mesoderm

progenitors in the chicken embryo[7]. Similar to our findings, an increasing pHi gradient is seen in murine colonic crypts[27,62]. However, reducing pHi in colonic crypts is associated with decreased expression of absorptive fate genes and increased Goblet cell and Paneth cell markers[27], which is in contrast to our findings in the small intestine. This discrepancy might be because of distinct differences between the small and large intestine, including the presence of protruded villi structure and Paneth cells, smaller crypts, more functional ISCs, and different transcriptional profiles[63]. Hence, the regulatory mechanism of ISC lineage responding to pHi dynamics in the small intestine may also be different. Comparing and contrasting the role of pHi in these two closely related organs will be an important topic for future study.

Our findings are consistent with roles for pHi in regulating WNT signaling and provide insight into the crosstalk between pHi dynamics and cell signaling. Specifically, decreased vacuolar pH promotes canonical WNT signaling by increasing LRP activation, and deceased cytoplasmic pH (pHi) regulating Disheveled localization[64], and β-catenin acetylation[7] and stability[56]. Thus, the pHi gradient in intestinal crypts may shape the pattern of WNT signaling, which is highest at the crypt base where pHi is low and decreases along the crypt axis where pHi is higher[65,66]. Given the complexity of WNT signaling and the heterogeneity of intestinal epithelial cells, quantitative determination of a pHi-WNT interplay along the crypt-villus axis is an important objective of future studies. In addition, our study provides evidence that pHi dynamics regulates cell-fate decisions downstream of a separate signaling cue, ATOH1, which is a master regulator of the secretory lineage. Regulation of lineage specification by pHi likely occurs within the progenitor cells that first upregulate ATOH1 to specify differentiation toward the secretory rather than absorptive fate. In addition, our observation that genes typically associated with the absorptive cell fate were upregulated in multiple *Atoh1*⁺ secretory cell types when pHi gradient is disrupted suggests that pHi dynamics continues to stabilize the secretory cell fate choice even after the initial lineage specification. Signaling by pHi dynamics is often mediated by intrinsic pH-sensing proteins with activity or protein-protein and protein-phospholipid binding affinities regulated within the cellular pH range[67]. Hence, selective targets of ATOH1 that are required for secretory lineage specification may be pH sensors with increased pHi enabling their transcriptional activity. Alternatively, other pHi-sensitive proteins may be acting as co-factors of ATOH1 to selectively regulate the expression of secretory fate determination genes.

Our findings with disrupting the pHi gradient are consistent with recent studies of plasticity of small intestinal epithelium, further demonstrating the flexibility of pHi as a regulator for cell behaviors. Specifically, our finding that with loss of NHE1 activity there is an increase in the number of *Clu*⁺ cells, which is a signature of intestinal regeneration, suggests that inhibiting NHE1 activity or decreasing pHi may elicit a response that mimics tissue regeneration induced by

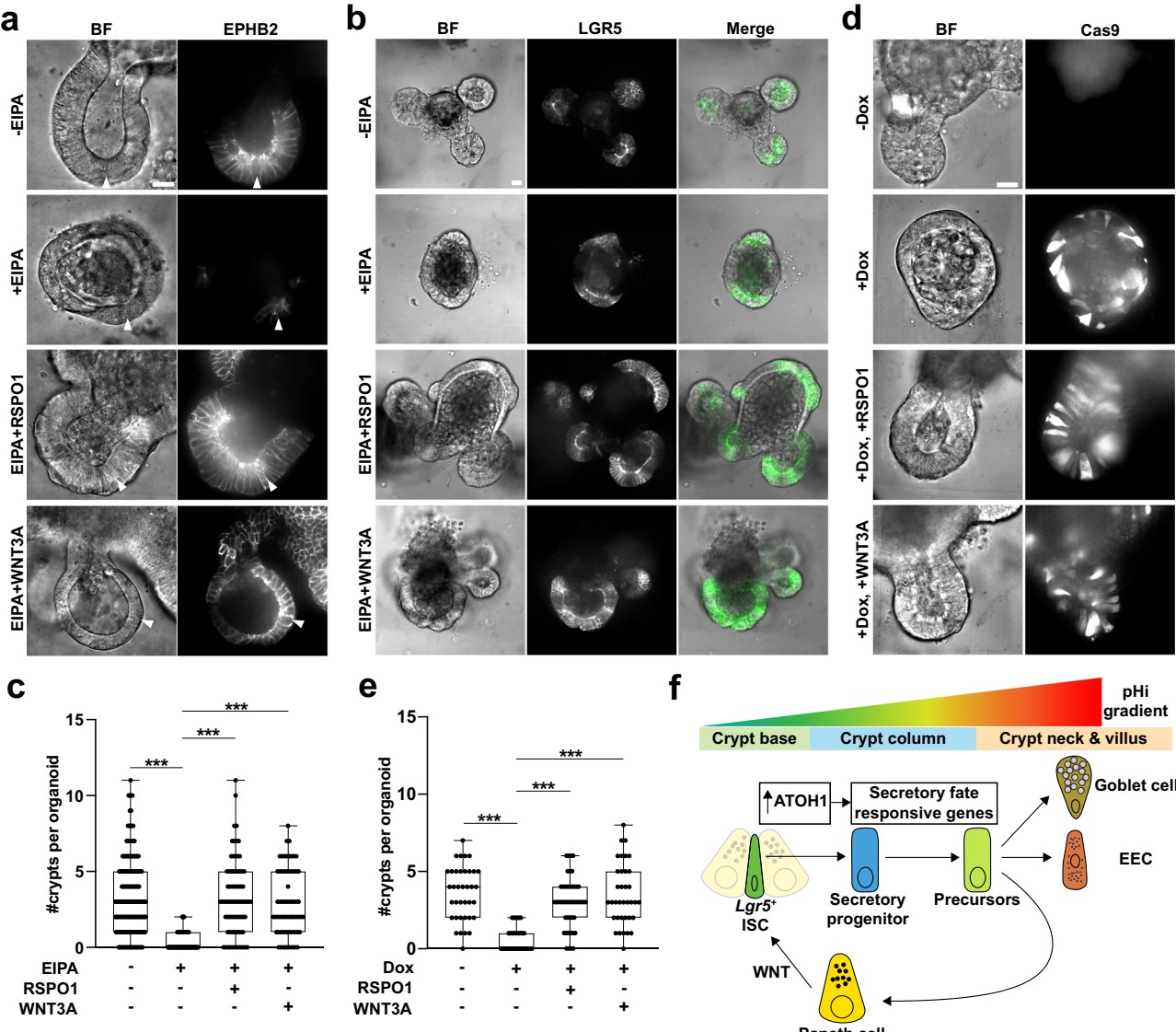

**Fig. 9 | Exogenous WNT rescues crypt budding impaired with loss of NHE1 activity. a** Exogenous WNT restores EPHB2 impaired by NHE1 inhibition. Representative confocal images from 3 cell preparations of EPHB2 immunolabeled organoids in the absence of EIPA (control), the presence of EIPA, or with EIPA plus increased Rspondin1 or exogenous WNT3A. BF, brightfield; arrowheads, Paneth cells in a crypt region. Quantified EPHB2 immunolabeling (see Supplementary Fig. 10b). **b–e** Representative confocal images and quantification of budded crypts at day 3 in $Lgr5^{DTR\text{-}GFP}$ organoids with NHE1 inhibition in the presence of exogenous WNT. Crypt growth in (**b, c**) the WT organoids in the absence (control) and presence of EIPA, and in (**d, e**) the inducible CRISPR-Cas9 organoids without (control) and with Dox, and supplemented with increased RSPO1 or WNT3A. BF, bright-field

view; LGR5, $Lgr5^+$ ISC. Data are from $n = 5$ biologically independent experiments in the WT organoids, and $n = 3$ biologically independent experiments with the inducible CRISPR-Cas9 organoids. All data are statistically analyzed by the two-sided Mann–Whitney test. All box plots are minimum to maximum, the dashed line shows 25th-75th percentiles, and the median is indicated as a central line. Data in all box plots with a statistically significant difference: *$p < 0.05$, **$p < 0.01$, ***$p < 0.001$, otherwise are not significantly different. $p$ in (**c**): $^{\text{-EIPAvs+EIPA}}p < 2.2\text{E-16}$; $^{\text{+EIPA\_RSPO1vs+RSPO1}}p = 4.265\text{E-16}$; $^{\text{+EIPA\_WNT3Avs+WNT3A}}p = 6.157\text{E-16}$. $p$ in (**e**): $^{\text{-Doxvs+Dox}}p = 1.365\text{E-09}$; $^{\text{+Dox\_RSPO1vs+RSPO1}}p = 2.148\text{E-08}$; $^{\text{+Dox\_WNT3Avs+WNT3A}}p = 6.236\text{E-09}$. Source data are provided as a Source Data file. All scale bars, 20 μm. **f** Schematic representation of the main findings.

intestine injury in vivo[14,17,42,43]. Hence, targeting NHE1 activity may be an alternative approach to current methods such as the use of radiation for investigating acute tissue regeneration in the intestinal epithelium. In addition, our studies are consistent with current views that crypt budding is maintained by a constant pool of ISCs[45,68], which is in line with our findings here, or by actin remodeling that drives a change in mechanical forces[30,69,70], which has been shown to be regulated by pHi in other contexts[71–73].

Our findings also have clinical relevance. Cystic fibrosis (CF) is associated with increased intestinal epithelial cell proliferation and an increased risk for gastrointestinal tumors. Increased pHi is seen with CF, and recent reports indicate that impaired intestinal homeostasis

with CF is in part mediated by disrupted pHi dynamics[74,75], including a dysregulated pool of ISCs and Paneth cells, as we found. Additionally, constitutively increased pHi is seen in most cancers[5,76], which could promote expansion of $Lgr5^+$ stem cells critical for initiating gastrointestinal cancers[13,77,78]. Consistent with this possibility, decreasing pHi by genetic silencing of NHE1 or carbonic anhydrase 9 significantly reduces the growth of colon cancer cells[79]. Additionally, most colorectal cancers are caused by mutations in key components of the WNT pathway, which would avert pHi regulated WNT pathway activity. Our findings on pHi regulating ATOH1-dependent ISC lineage specification that connects ISC maintenance and the WNT circuit (Fig. 9f) adds insight into the development of CF and intestinal tumorigenesis and

provides a biological basis for targeting NHE1 and pHi dynamics for potential therapeutics.

## Methods

This study complies with all relevant ethical regulations. Animal-use protocols were approved by the Institutional Animal Care and Use Committee (IACUC) of the University of California at San Francisco (AN197697-00 and AN192878-01G), and all experiments were conducted in accordance with the specified guidelines.

### DNA constructs and CRISPR design

The pLX304 lenti-mCherry-SEpHluorin plasmid was generated by inserting an mCherry-SEpHluorin fragment from an mCherry-SEpHluorin plasmid (Addgene #32001) into lentiviral vector pLX304 (Addgene #25890). The mCherry-SEpHluorin fragment was first amplified and then cloned into the Gateway entry vector pENTR/D-TOPO (Thermofisher) to build pENTR/D-mCherry-SEpHluorin. The mCherry-SEpHluorin fragment was then transferred from pENTR/D-mCherry-SEpHluorin into pLX304 via LR gateway reaction to build pLX304 lenti-mCherry-SEpHluorin. To generate the pTLCV2 NHE1 KO plasmid, guide RNA targeting the first exon of the mouse NHE1 gene was designed and cloned into an all-in-one doxycycline (Dox)-inducible vector TLCV2 (Addgene #87360). In brief, gRNA oligo forward (5′-caccgAACTTAATCATTGAACATGG-3′) and reverse (5′-aaacCCATGTTCAATGATTAAGTTc-3′) were first annealed and then ligated into BsmBI digested TLCV2 vector.

### Small intestinal organoids, mouse lines and culture

WT, *Lgr5*[DTR-GFP] organoids, *Lgr5*[CreER];*Rosa26*[RFP] organoids, and *Atoh1*[CreERT2];*Rosa26*[tdTomato] organoids stocks were originally obtained from 10 weeks old adult mice. The mCherry-SEpHluorin and inducible NHE1 CRISPR-Cas9 KO organoids were generated by infecting WT organoids with lentivirus made from pLX304 lenti-mCherry-SEpHluorin and pTLCV2 NHE1 KO plasmids, respectively, followed by antibiotic selection using 10 μg/mL blasticidin for mCherry-SEpHluorin and 2 μg/mL puromycin for NHE1 CRISPR-Cas9 organoids, as described[79]. Single clones of NHE1 silenced organoids were established from selected individual puromycin-resistant organoids. The C57BL/6 J (male, 10 weeks old) and mTmG (female, 10 weeks old) adult mice were used for isolating fresh crypts for ex vivo pHi measurement. All mice were housed at a constant temperature (18–23 °C) and a controlled light cycle (12 h light/dark), with food and water provided ad libitum. The organoid cultures were generated and maintained in a 3D format[13,79,80]. Briefly, organoids were embedded in Matrigel (Corning, 356231) and cultured in a standard 24-well plate. For confocal imaging, organoids embedded in Matrigel were seeded into 24-well glass-bottom MatTek plates (MatTek, P24G-0-10-F). Unless stated otherwise, organoids were grown in ENR medium containing advanced DMEM/F12 (Invitrogen, 12634-028) supplemented with 50 ng/mL EGF (Sigma-Aldrich, E9644-.2MG), 100 ng/mL Noggin (R&D, 6057-NG/CF), R-spondin1(RSPO1) (conditioned medium, CM; CCHMC RT0457, 1.6% v/v), 10 mM HEPES (Invitrogen, 15630-080), 1 mM N-acetylcysteine (Sigma-Aldrich, A7250), 1X glutaMAX (Invitrogen, 35050-061), 1X N2 supplement (Invitrogen, 17502-048), 1X B27 supplement (Invitrogen, 17504-044), and 1X penicillin-streptomycin (pen/strep). To induce spheroid formation, organoids were cultured in ENRWN medium, which includes ENR medium with 50% v/v WNT3A (CM, ATCC CRL-2647) and 10 mM Nicotinamide (Sigma-Aldrich, N1630-100MG). For post-infection cultures, medium with an additional 10 μM of the Rho kinase inhibitor Y-27632 (Sigma-Aldrich, Y0503-1MG) in ENRWN was used for recovery. For WNT rescue, ENR medium was supplemented with 6.4% v/v RSPO1 (CM, CCHMC RT0457) or 20% v/v WNT3A (CM, ATCC CRL-2647). For single-cell reassociation assays, ISC-Paneth pairs were grown in single-cell growth medium, which was ENR with 10 mM Nicotinamide, 2.5 μM

Y-27632, 2.5 μM Chir99021 (Sigma-Aldrich, SML1046-5MG), 1 μM Jagged-1 (Anaspec, AS-61298), 2.5 μM Thiazovivin (Selleckchem, S1459).

### NHE1 CRISPR-Cas9 silencing and validation in organoids

To induce and validate NHE1 CRISPR-Cas9 editing, 2 μg/mL doxycycline (Dox) was added to the ENR medium and to Matrigel after passaging organoids. After induction for 2–3 days, clones containing at least 80% Cas9-EGFP positive organoids were selected for genomic DNA (gDNA) extraction. The gDNA was isolated from selected clonal organoids using the gDNA tissue miniprep system (Promega, A2051) and evaluated by PCR and DNA gel electrophoresis using primer pair forward (5′- GCCCGTGGTCCAGCCTATC-3′) and reverse (5′-GTCCCATCCCAGCTGTAGGAGA-3′). Purified PCR fragment containing the edited DNA was sequenced by Sanger sequencing using primer forward (5′-CGTCTGGGGATTTCATCCACCT-3′) and reverse (5′- CTATCTTCATGAGGCAGGCCAGGA-3′), respectively. The sequencing results were analyzed by comparing WT and -Dox samples (Supplementary Fig. 2).

### pHi measurement in organoids

Ratiometric imaging of the pHi biosensor mCherry-SEpHluorin was performed as described[18]. To inhibit NHE1 activity, 5 and 10 μM EIPA were tested with WT organoids. The 5 μM EIPA was selected based on its efficacy in blocking crypt budding and its effect on epithelium viability. As the crypt development took 3 days, day 1 and day 3 were selected for capturing crypt budding. In brief, organoids embedded in Matrigel were cultured in ENR medium supplemented with DMSO vehicle (1:1000) or 5 μM EIPA for 3 days, and a microscopy-based measurement was performed on day 1 and day 3 organoids. Before imaging, growth medium was replaced with freshly made pHi buffer containing 25 mM NaHCO$_3$, 115 mM NaCl, 5 mM KCl, 10 mM glucose, 1 mM K$_3$PO$_4$, 1 mM MgSO$_4$, and 2 mM CaCl$_2$ pH 7.4 and including the membrane dye CellMask™ (Molecular probes, C10046) (1:1000 dilution) for 10 min. After washing with fresh pHi buffer not containing CellMask™ fluorescence images were acquired using a customized spinning disk confocal (Yokogawa CSU-X1) on a Nikon Ti-E microscope equipped with a live-cell imaging chamber maintained with 5% CO$_2$ at 37 °C, a 40X water objective, 488 nm, 560 nm, and 590 nm excitation lasers. and a Photometrics cMYO cooled CCD camera. Images were stored as Z-stacks of 2 μm-optical sections. To calibrate fluorescence ratios of mCherry-SEpHluorin to pHi, after each experiment, organoids were incubated for 15-20 min with a KCl buffer (80 mM KCl, 50 mM K$_3$PO$_4$, 1 mM MgCl$_2$) containing 20 μM nigericin (Invitrogen, N1495) at pH 7.8. After acquiring fluorescence ratios, organoids were washed and incubated with nigericin buffer at pH 6.6 and fluorescence ratios were again acquired to perform a 2-point calibration conversion.

For image analysis, backgrounds were removed from Z stacks and single optical sections containing representative intestinal crypts were chosen for quantification. Individual cells were segmented by CellMask™ dye. To identify major cell types in the crypt, Paneth cells were selected based on size and presence of visible intracellular granules under both bright-field and mCherry channels, ISCs were identified based on their location which is intermingled between two Paneth cells, other cells along the crypt column starting at a position two-cell above the upmost Paneth cell and ending at the crypt neck, the upmost part of a budded crypt, were further specified as the crypt column and the crypt neck. In the crypt without a bud, up to four of the cells closest to the upmost Paneth cell were chosen as the column for pHi measurement. Because the crypt region and villus region were less distinguishable by live-cell imaging in the day 1 organoid during development, the column region was not specified for pHi determination on day 1. To calculate mCherry/SEpHluorin intensity ratios, fluorescence intensity was measured in the selected individual Paneth cells, ISCs, and cells located in the column and neck using ImageJ. In Microsoft Excel, the pHi values were determined by applying those intensity ratios to a standard curve generated by graphing the nigericin

pH values (Y-axis) against their dependent intensity ratios (X-axis). As the villus thickened and enlarged as a spheroid, background signals started to mask the actual fluorescence from mCherry and SEpHluorin. This led to an unreliable mCherry/SEpHluorin ratio and resulted in a higher-than-normal pHi value (>7.8) that was out of the linear range of SEpHluorin (6.7-7.7). Therefore, pHi in the villus cells was not reported.

## pHi measurement in isolated crypts

Small intestinal crypts were freshly isolated from adult old mice[10]. Isolated crypts from C57BL/6 J (male, 10 weeks old) and mTmG (female, 10 weeks old) mice were washed and suspended in a basal medium containing advanced DMEM/F12 (Invitrogen, 12634-028) supplemented with 50 ng/mL 10 mM HEPES (Invitrogen, 15630-080), 1 mM N-acetylcysteine (Sigma-Aldrich, A7250), 1X glutaMAX (Invitrogen, 35050-061), and 1X penicillin-streptomycin (pen/strep). A pHi-sensitive dye BCECF (Invitrogen, B1170) was added in the suspension at final concentration of 10 mM. After a 15 min-incubation at 37 °C, crypts were washed with basal medium and loaded on coverslips for microscopy. Images were acquired at Ex 440 nm/ Em 530 nm (pH insensitive) and Ex 490 nm/ Em 530 (pH sensitive) using a 20x air lens on a Leica SP5 laser scanning confocal microscope. To determine pHi values in a crypt, crypt bottom was identified first by a presence of Paneth cells. Then, in a bottom-up manner, individual crypts were divided into four sections (bottom, lower column, higher column, and top) with equal length guided using the line tool in Image J. The pHi values were determined by calibrating the intensity ratio of pH sensitive and pH insensitive channels from 5 ROIs in each section using the same method for the mCherry-SEpHluorin reporter.

## Crypt budding quantification and imaging

Under a bright-field view on the spinning disc confocal microscope, a budded crypt was defined by the epithelial protrusion (containing visible Paneth cells) grown out of the spherical cyst with the extension protruded larger than the thickness of the epithelial layer of the cyst. To quantify crypt budding, the number of budded crypts were counted in organoids under each of the following conditions. (1) For -EIPA vs +EIPA conditions, day 3 organoids grown in ENR medium supplemented with DMSO (vehicle control) or 5 μM EIPA were scored for crypt budding. (2) For inducible CRISPR-Cas9 mediated NHE1 silencing, organoids in ENR medium were treated with or without Dox for 2 days and then re-seeded in ENR with or without Dox for another 3 days before quantification. (3) For WNT rescue experiments, EIPA-treated and CRISPR-Cas9 silenced organoids were cultured in the ENR medium supplemented with either 3X RSPO1 or 20% exogenous WNT3A for 3 days prior to quantification. To imaging crypt budding in the $Lgr5^{DTR-GFP}$ organoids (Fig. 2a), a 3-day time-lapse imaging (bright-field, FITC; 40X) began immediately after seeding (day 0) in ENR + DMSO (vehicle control) and ENR + 5 μM EIPA. Images were acquired as Z-stacks of 2 μm optical sections every 3 h. To image pHi dynamics in crypt budding in the mCherry-SEpHluorin organoids, a time course was acquired (bright-field, FITC, PE-Texas red; 20X) was starting one day after seeding, in which a 2 μm Z-stack image set was taken every 30 min for two days.

## Single-cell RNA-sequencing (scRNA-seq) of intestinal organoids

To obtain single-cell suspension from the organoids without or with NHE1 inhibition (± EIPA, ± Dox), organoid cultures were first washed with the cold (4 °C) advanced DMEM/F12 (Invitrogen, 12634-028) supplemented with 10 mM HEPES (Invitrogen, 15630-080), 1 mM N-acetylcysteine (Sigma-Aldrich, A7250), 1X glutaMAX (Invitrogen, 35050-061), and 1X pen/strep. After the washing, wells containing the organoids were incubated with 300 uL cold (4 °C) Corning recovery solution (Corning, 354253) for 15 min to melt the Matrigel. Next, organoids were mechanically disrupted with the P1000 pipette tips first and then trypsinized with RT TrypLE Express (Gibco, 1952062) for

up to 15 min to disassociate into single cells. Single cells were then suspended within a sorting buffer that contained Hanks' balanced salt solution (HBSS) with 3% m/v FBS, 10 mM HEPES and 5 mM EDTA. After staining with Live/Dead (Invitrogen, L23105, 1:1000) in the sorting buffer, fluorescence-activated cell sorting (FACS) was performed on the BD FACSAira instrument (FACSDiva 8.0.1) to enrich either live cells (±EIPA and -Dox conditions) or live GFP+ cells (+Dox condition) (Supplementary Fig.3). Approximately 100,000-150,000 cells per genotype/condition were collected and re-suspended in ~200 μL 1X PBS containing 0.04 % (w/v) BSA (Ambion, AM2616). Purity was more than 90% and determined by flow cytometry using the Live/Dead (Invitrogen, L23105, 1:1000) stain. For library construction and sequencing, approximately 20,000 cells per genotype/condition were loaded onto a 10X Genomics Chromium Next GEM chip to generate a single cell 3′ gene expression library using the 10X Chromium Next GEM Single Cell 3′ regent kits v3.1. Pooled libraries were sequenced (paired-end, single indexing) on an Illumina NovaSeq 6000 sequencer with a standard sequencing protocol (Read 1, 28 cycles; i7 Index, 8 cycles; i5 Index, 0 cycles; Read 2, 91 cycles). The raw sequence outputs were filtered and aligned (mouse reference, mm10) to produce feature-barcode matrices using the Cell Ranger pipeline 6.1.2 on 10X Genomics Cloud Analysis.

## Bioinformatic analysis

Single-cell feature-barcode matrices were analyzed using Seurat 4.1 on RStudio server 2022.01.999[81]. We attained an averaged number of 23789 UMIs and 17882 genes (averaged median of 2282 genes detected per cell) in the WT control (-EIPA), and 19777 UMIs and 17291 genes (averaged median of 2575 genes detected per cell) in EIPA-treated organoids from 2 separate preparations. We also obtained 22053 UMIs and 17291genes (median of 2942 genes detected per cell) in CRISPR-Cas9 control (-DOX), and 14,314 UMIs and 17,630 genes (median of 3507 genes detected per cell) in CRISPR-Cas9 silenced (+DOX) organoids from 1 preparation. For the pre-processing[34], we selected cells based on the QC metrics (unique genes, total number of RNAs, and mitochondrial RNA) for each genotype/condition (Supplementary Fig. 4a). After QC filtration, we combined all datasets and then performed normalization and variance stabilization via sctransform[82]. After data integration, principal component analysis (PCA) was used to reduce the dimensionality of the combined dataset followed by constructing a shared nearest neighbor (SNN) graph (dimensions of reduction = 1:45, resolution =0.8) and building clusters with uniform manifold approximation and projection (UMAP) dimensional reduction (dimensions of reduction =1:45). To set the identities of clusters in the combined dataset, expression comparison was performed on the integrated data. In brief, markers that were distinctively enriched in each cluster were first calculated and displayed for general identification. Then, we further modified the clustering based on the average gene expression profiles of well-characterize cell-type markers. To enrich the progenitors and precursors in the secretory lineage, we subclustered a parental cluster associated with elevated secretory genes using the same PCA-SNN-UMAP workflow above (dimensions of reduction = 1:13, resolution =1), followed by subcluster identification using signature gene average expression. Lastly, we assigned each cluster with a cell-type identity in the combined dataset. A "unknown proliferative" cluster was included due to the enrichment of several minichromosome maintenance (Mcm) genes but without a specific cell-type identification. For downstream gene analysis, a log scale normalization (log1p scale; scale.fatctor = 10,000) and a data scaling (variables to regress out, total number of RNAs and mitochondrial RNA) were performed on the identity-assigned dataset. To quantify the relative cell number change with NHE1 inhibition, we first calculated a relative cell count of the WT control (-EIPA) and EIPA-treated clusters by calculating the number of cells in the cluster divided by the total cell count in the dataset. We then calculated a relative change by dividing

the relative cell count in the EIPA-treated dataset by that of the WT control. We did not perform this calculation for the CRISPR datasets because we needed to enrich for GFP⁺ (Cas9⁺) cells to obtain the +Dox dataset, and this may have introduced a bias in the types of cells collected. To compare any gene expression between conditions (±EIPA or ±Dox) in different cell types, all calculations and plotting were performed using the normalized/scaled RNA data unless stated otherwise.

## Proliferation assay and crypt cell quantification

Click-iT Edu cell proliferation kit (Invitrogen, C10340) and Ki67 labeling were used to assess crypt proliferation. For the Edu assay, day 3 WT organoid culture was first incubated in 10 µM Edu solution for 2 h then proceeded for immunostaining. To quantify the proliferative cells within CD44⁺ clusters, numbers of Edu⁺ and CD44⁺ positive cells in each CD44 labeled cluster were first counted separately from confocal z-plane sectioning through the middle of a budded crypt or an unbudded organoid. For normalization, a ratio of numbers of Edu⁺,CD44⁺ double-positive cells to the numbers of total CD44⁺ cells were then calculated as a measurement of overall proliferation. For the Ki67 assay, day 3 *Lgr5*^DTR-GFP organoids were fixed and immunolabelled with Ki67 antibody (1:250). To quantify proliferative ISCs in the *Lgr5*^DTR-GFP organoids, the percent numbers of proliferative ISCs per crypt were measured by a ratio of the numbers of Ki67⁺,LGR5⁺ cells to the total numbers of LGR5⁺ cells in a representative optical section of a crypt region indicated by LGR5. The cell number was determined by Hoechst staining of crypt regions defined by GFP⁺ cells. To quantify other crypt cells, the numbers of *Lgr5*⁺ ISCs and *Lgr5*⁻,CD44⁺ cells (crypt progenitors) within each CD44⁺ cluster were counted and normalized to the total number of CD44⁺ cells, in the same manner as the proliferation assay. To count Paneth cells in developing organoids, organoids were fixed at day1, day2, and day3 for immunostaining. All LYZ⁺ Paneth cells present in each CD44⁺ cluster were counted in 3D confocal images and normalized to the arc length of the 3D projection of CD44⁺ clusters.

## Immunolabeling

Matrigel embedded organoids were fixed in 2% paraformaldehyde for 45 min at RT. Fixed organoids were washed 3 × 20 min at RT with 1X Glycine-PBS (7.5% m/v) followed by washing 2 × 10 min with PBS. Organoids were then permeabilized in 0.5% Triton X-100 for 15 min at RT and blocked in PBS buffer containing 0.1% BSA, 0.2% (v/v) Triton X-100, and 0.04% (v/v) Tween-20, 5–10% (v/v) serum overnight at 4 °C. Organoids were incubated overnight at 4 °C with primary antibodies diluted in blocking buffer. Antibodies and dilutions included anti-CD44 (BioLegend, 103030, 1:500), anti-lysozyme (LYZ) (Dako, EC 3.2.1.17, 1:100), anti-EPHB2 (R&D, AF467, 1:25), anti-DLL1(R&D, AF5026. 1:14), anti-Aldolase B (ALDOB; Abcam, ab75751, 1:200), and anti-Chromogranin A (CHGA) (1:250). After washing with blocking buffer (w/o serum) 3 × 1 h at RT, organoids were incubated overnight at 4 °C with fluorescent secondary antibodies (Molecular Probes) diluted 1:200 in blocking buffer. After incubation, organoids were washed 3 × 1 h with blocking buffer (w/o serum) and then 20 min with PBS. Washed organoids were stained with Hoechst 33342 (Molecular probes, 1:1000) and stored in PBS at 4 °C. Fluorescence images were acquired using a customized spinning disk confocal (Yokogawa CSU-X1) on a Nikon Ti-E microscope with a 40X water objective equipped with a Photometrics cMYO cooled CCD camera.

For image analysis, background of the region of interest in each section of Z-stacks was corrected by subtracting a value of mean intensity from a sample-free region and single optical sections containing representative intestinal crypts were chosen for quantification. Individual cells were segmented by the membrane dye. To identify major cell types in the crypt, Paneth cells were selected based on size and presence of visible intracellular granules under both bright-field and mCherry channels, ISCs were identified based on their location

which is intermingled between two Paneth cells, other cells along the crypt column starting at a position two-cell above the upmost Paneth cell and ending at the crypt neck, the upmost part of a budded crypt, were further specified as the crypt column and the crypt neck. In the crypt without a bud, up to four of the cells closest to the upmost Paneth cell were chosen as the column for pHi measurement. Because the crypt region and villus region were less distinguishable by live-cell imaging in the day 1 organoid during development, the column region was not specified for pHi determination on day 1. To calculate mCherry/SEpHluorin intensity ratios, fluorescence intensity was measured in the selected individual Paneth cells, ISCs, and cells located in the column and neck using ImageJ. In Microsoft Excel, the pHi values were determined by applying those intensity ratios to a standard curve generated by graphing the nigericin pH values (Y-axis) against their dependent intensity ratios (X-axis). Because of the mCherry/SEpHluorin intensity ratio in the cells in the organoid villus (above the crypt neck) was too high to be calibrated, the pHi in those cells were not determined.

## Single cell reassociation assay

NHE1 silencing was induced by adding 2 µg/mL Dox to the inducible-NHE1 CRISPR Cas9-GFP organoids culture for 2 days. On the day of sorting (Supplementary Fig. 9), the NHE1 CRISPR silenced organoids and *Lgr5*^DTR-GFP organoids were first dissociated and then stained for CD44 (BioLegend, 103030, 1:250), CD24 (BD, 553262, 1:250), and Live/Dead (Invitrogen, L23105, 1:1000) in the sorting buffer. FACS was performed on the BD FACSAria (FACSDiva 8.0.1) with a purity more than 90%. Live *Lgr5*⁺ ISCs (*Lgr5*-DTR-GFP^high, CD44^high, CD24^low, and side-scatter^low) were sorted (Supplementary Fig. 9a) and mixed with either the sorted live WT Paneth cells (*Lgr5*-DTR-GFP^negative, CD44^high, CD24^high, and side-scatter^high) (Supplementary Fig. 9b) or the sorted live NHE1-silenced Paneth cells (Cas9-GFP^high, CD44^high, CD24^high, and side-scatter^high)[30] (Supplementary Fig. 9c) at 1:1 ratio (400-500 cells each). In addition, groups containing the *Lgr5*⁺ ISCs alone, the *Lgr5*⁺ ISCs + WNT3A (20% v/v WNT3A in single-cell growth media), the WT Paneth cells alone, and the NHE1-silenced Paneth cells alone were also included as controls for normalization. All groups were grown in single cell growth media for 3 days and then switched to normal ENR media. To evaluate organoid formation ability from the ISC-Paneth cell pairs, the number of organoids formed in all the groups (if any) was counted on day 8 and normalized to the *Lgr5*⁺ ISCs + WNT3A group.

## Lineage tracing

To reveal overall ATOH1⁺ secretory cell production, day 1 control and EIPA-treated *Atoh1*^CreERT2; *Rosa26*^tdTomato organoid culture were added with 0.1 µM TAM for 24 h first (no washout) before starting a 48 h time-lapse imaging (day 2 to day 3), which were acquired at 30-min intervals, in a 2-µm z step size per optical section. For measuring the overall secretory cell production, changes of newly produced ATOH1⁺ cells (RFP⁺) numbers from day2-3 were normalized to the number of ATOH1⁺ cells seen on day 2. For other lineage trancing experiments, the *Atoh1*^CreERT2; *Rosa26*^tdTomato and the *Lgr5*^CreER; *Rosa26*^RFP organoids were prepared by treating the day 1 organoids with 0.1 µM 4-hydroxytamoxifen (TAM; Sigma-Aldrich, 579002) for 26 h followed by washing and passaging. Reseeded organoids were cultured with vehicle control or 5 µM EIPA for 2–3 days either in a standard culture incubator before immunostaining for DLL1 and ALDOB, or in a live-cell imaging chamber for confocal microscopy for Paneth cell production, recorded every 24 h, with z-stacks of 2 µm optical sections covering the entire crypt region. To quantify lineage tracing, numbers of RFP⁺ or tdTomato⁺ cells were counted in the crypt regions, indicated by immunolabelled CD44⁺ clusters in the *Lgr5*^CreER; *Rosa26*^RFP organoids, or by clusters of live ATOH1⁺ cells and Paneth cells in the *Atoh1*^CreERT2; *Rosa26*^tdTomato organoids. To assess the Paneth cell production, changes of newly generated ATOH1⁺ Paneth cells (RFP⁺/dense granule⁺)

numbers within 24 h were normalized to the number of total ATOH1+ cells (RFP+) as well as the Atoh1+ non-Paneth cells (RFP+/dense granule-). To determine differentiation capability of *Lgr5*+ ISCs into DLL1+ enriched secretory progenitors, the Paneth cells, and the enterocytes, percent of crypts containing RFP+/CD44+/DLL1+ (secretory progenitors), RFP+/CD44+/LYZ+(Paneth cells), and organoids containing RFP+/CD44-/ALDOB+ cells were calculated. The number of RFP+/CD44+/DLL1+, RFP+/CD44+/LYZ+, RFP+/CD44-/ALDOB+, and RFP+ cells were also quantified in each crypt or organoid as a measurement for *Lgr5*+ ISC differentiation.

## RNA extraction and quantitative reverse transcription PCR (qRT-PCR)

Total RNA was extracted from organoids using the RNeasy plus mini kit (Qiagen, 74134). RNA was used to generate cDNA using the iScript cDNA synthesis kit (Bio-Rad, 1708890). The mRNA level of target genes was accessed by qRT-PCR on a Real-time PCR system (Applied Biosystems) using the SYBR green supermix reaction (Bio-Rad, 170-8882). The relative expression of targets was analyzed using the double delta $C_t$ method by normalizing to *Gapdh*. Primer sequences were below: *Lgr5* forward (5'-GGGAGCGTTCACGGGCCTTC-3'), *Lgr5* reverse (5'-GGTTGGCATCTAGGCGCAGGG-3'); *Lyz* forward (5'-GGAATGGATGGC-TACCGTGG-3'), *Lyz* reverse (5'-CATGCCACCCATGCTCGAAT-3'); *Alpi1* forward (5'-AGGATCCATCTGTCCTTTGG-3'), *Alpi1* reverse (5'-ACGT TGTATGTCTTGGACAG-3'); *ChgA* forward (5'-CTCGTCCACTCTTT CCGCAC-3'), *ChgA* reverse (5'-CTGGGTTTGGACAGCGAGTC-3'); *Muc2* forward (5'-ATGCCCACCTCCTCAAAGAC-3'), *Muc2* reverse (5'-GTAGTTTCCGTTGGAACAGTGAA-3'); *Hes1* forward (5'-GCTCACTTCG-GACTCCATGTG-3'), *Hes1* reverse (5'-GCTAGGGACTTTACGGGTAGCA-3'); *Atoh1* forward (5'-GCCTTGCCGGACTCGCTTCTC-3'), *Atoh1* reverse (5'-TCTGTGCCATCATCGCTGTTAGGG-3'); *Axin2* forward (5'-AGTGTCT CTACCTCATTTTCCG-3'), *Axin2* reverse (5'-CTTTCCAGCTCCAGTTT-CAGT-3'); *Ascl2* forward (5'-GCTGCTTGACTTTTCCAGTTG-3'), *Ascl2* reverse (5'-CACTAGACA GCATGGGTAAGG-3'); *Olfm4* forward (5'-TGAAGGAGATGCAAAAACTGG-3'), *Olfm4* reverse (5'-CTCCAGCTTC TCTACCAAGAGG-3'); *Gapdh* forward (5'-TGTGTCCGTCGTGGATC TGA-3'), *Gapdh* reverse (5'-CCTGCTTCACCACCTTCTTGA-3').

## Statistics & reproducibility

For scRNA-seq datasets, the statistical analysis was performed on RStudio using the method Differential Expression with Capture Efficiency adjustment (DECENT)[83,84]. For other experimental data, the statistical analysis was performed using GraphPad Prism 8 and RStudio. In GraphPad Prism 8, outliers were determined using the ROUT method by fitting data with nonlinear regression before additional analysis. Data fitting a normal distribution were analyzed using two-sided tests, including normal student's *t* test, Welch's *t* test, or Tukey-Kramer test. The non-normally distributed data were analyzed by the two-sided Mann–Whitney ranking test or Wilcoxon Signed Rank test. No statistical method was used to predetermine sample size. As the organoid culture used in each experiment intrinsically consists of ten to hundreds of individual organoids, in general, at least 80% of all available organoids in the culture was randomly sampled for each experiment. No data were excluded from the analyses. The investigators were not blinded to allocation during experiments and outcome assessment (partial blinding was involved in data analysis).

## Reporting summary

Further information on research design is available in the Nature Portfolio Reporting Summary linked to this article.

## Data availability

Source data are provided with this paper. The single-cell RNA sequencing data generated in this study have been deposited in NCBI gene expression omnibus under accession code GSE211097. Source data are provided with this paper.

## Code availability

Code for bioinformatic analysis is deposited in a GitHub repository and is freely accessible at https://zenodo.org/record/7922817 [https://zenodo.org/record/7922816].

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

## Acknowledgements

We thank past and present members of the Barber and Nystul groups for insightful discussions and support throughout the project. We appreciate the help from the Wittmann lab (UCSF), the Kutys lab (UCSF), and the Conti lab (USCF), particularly Dr. Torsten Wittmann, Dr. Jefferey van Haren, and Dr. Dema Alessandro for help with confocal microscopy, and Dr. Marco Conti and Dr. Peter Althoff for providing a mouse strain. We thank UCSF Parnassus Flow CoLab (RRID: SCR_018206) for the assistance on FACS. This study is supported by National Institutes of Health grant (CA197855 D.L.B.), National Science Foundation grants (P0538109 T.N., D.L.B., 1933240 T.N.), UCSF Research Allocation Program, UCSF RAP Pilot for Established Investigators in Basic and Clinical/Translational Sciences (T.N.), National Institutes of Health, Maximizing Investigators' Research Award (DE026602 O.D.K.), National Institutes of Health, Shared equipment grant (S10OD028611-01).

## Author contributions

Project conceptualization was performed by Y.L., D.L.B., and T.N. Experimental design was developed by Y.L., D.L.B., T.N., E.R., D.C., and O.D.K.. Y.L. carried out the investigation that was supervised by D.L.B. and T.N.. The original manuscript draft was written by Y.L., T.N., and D.L.B, and was reviewed by Y.L., T.N., D.L.B, E.R., D.C., and O.D.K..

## Competing interests

The authors declare no competing interests.

## Additional information

**Peer review information** : *Nature Communications* thanks the anonymous reviewer(s) for their contribution to the peer review of this work. A peer review file is available.

