## [Peer Review File · Nature Communications]

Intracellular pH dynamics regulates intestinal stem cell lineage specificationREVIEWER COMMENTS

Reviewer #1 (Remarks to the Author):

The manuscript of Liu and co-workers 'Intracellular pH dynamics regulates intestinal stem cell lineage specification' reports the differences ('gradient across crypt axis') in pHi between crypt and villae-localised intestinal cells in the mouse intestinal organoid model. This finding is novel and is based on the use of genetically encoded ratiometric fluorescence pH-sensitive biosensor protein coupled with the use pharmacological inhibition of Na⁺/H⁺-exchanger. Authors show that this finding is important for the Paneth cell function and regulates differentiation of intestinal stem cells either towards absorptive or secretory lineages. I believe that this work is important for the the interdisciplinary fields of organoids, function of the intestinal stem cells, single-cell imaging and dynamic multi-parameter live fluorescence microscopies.

I have only few comments, which however may require performing some additional experiments and revision of the manuscript:

Major points:

1. It was recently shown that intestinal stem cells display rather dynamic metabolism, which can be strongly dependent on the availability of the metabolic substrates in the medium - glucose, pyruvate and perhaps others (Okkelman et al, Redox Biology, 2020, 30, 101420) and thus it is important to see if e.g. glucose present in the medium would affect the observed pHi gradients. This could happen if e.g. stem cells activate their OxPhos. Therefore I would be interested to see if the observed crypt-villi gradient of pHi is (not) dependent on cell metabolism and availability of the glucose in the medium. I see that authors used buffer supplemented with 10 mM Glc (with bicarbonate, 115 mM NaCl etc.) in the experiments but it also does not contain pyruvate, glutamine amino acids etc. How long the incubations were performed? Do you think that pHi in the intestinal crypt cells would change with duration of incubation in the experimental buffer? Please discuss and provide experimental evidence, if possible.
2. Can the inhibition of NHE1 be not 100% specific with EIPA? Please discuss or perhaps add experimental evidence e.g. with dimethyl amiloride or other inhibitor, e.g. from the article: <https://bpspubs.onlinelibrary.wiley.com/doi/full/10.1111/bph.14429>

Minor points:

1. Page 12, "for the Paneth cell marker, LYZ" - add reference for lysozyme as a marker.
2. Page 21. "5 uM EIPA for 1 to 3 days." - how this was optimised? Perhaps I am missing it somewhere?
3. Page 21. Below, description of membrane dye CellMask. Please provide used concentration.
4. Page 22. "segmented by the membrane dye." - which dye? CellMask?
5. Page 22. "Because of the mCherry/SEpHluorin intensity ratio in the cells in the organoid villus (above the..." - can you provide more specific values and perhaps S/N ratios?

Reviewer #2 (Remarks to the Author):

This paper by Liu, Reyes, Castillo-Azofeifa et al., report the existence of a crypt/villi axis intracellular pH gradient and uncovers the role of intracellular pH in the regulation of adult intestinal stem cell lineage specification in mouse small intestinal organoids. By using a pH biosensor, authors reveal the existence of an intracellular pH gradient along the crypt axis that is regulated by the activity of the Na⁺-H⁺ exchanger 1 (NHE1). Abolition of this gradient by inhibiting NHE1 with a selective pharmacological inhibitor (EIPA) or by doxycycline-

inducible CRISPR results in loss of crypts budding in intestinal organoids. By combining single-cell RNA sequencing and lineage tracing experiments, authors reveal that intracellular pH dynamics influence cell differentiation. Increased pH promotes secretory cell fate while reduced pH favour differentiation into absorptive lineage.

This is an interesting study that reports a novel mechanism influencing tissue maintenance through stem cell differentiation in the intestine. However, additional work is required to fully support the conclusion of this study. There are many cases also where results described are not reflected in the information presented in figures.

Major points:

- This work is entirely carried out in vitro, using 3D cultured organoids. While this is a very valuable reductionist model, the main claims of this study, that there is an intracellular PH gradient in the crypt axis of the intestine that is functionally important for ISC differentiation need to be demonstrated in vivo to really confirm that the observations of this study are not influenced by culture conditions. For example, PH could be assessed in freshly dissected or fixed intestinal tissue samples. The functional role of NHE1 could be addressed by conditional gene knockout using Villin-Cre. I believe this will be the most robust way to confirm the results and to look at the impact of PH in intestinal cell differentiation.

- Figures 1B-1C: The conclusions made in the texts are not supported by the pictures shown in the figure. The authors conclude that there is a significantly lower pH in ISCs and adjacent non-Paneth cells when treated with EIPA compared to controls at day 3: this is difficult to see on the pictures shown in Figure 1B. It seems to be the contrary on the picture. Also the PH in Paneth cells at day three with and without EIPA seems to be non-significantly different as per the statistics shown in Figure 1C. It is also indicated that the pH remains unchanged in ISCs between controls and EIPA-treated organoids at day 1. On the picture shown in Figure 1B, it seems to be the contrary: the pH seems to be higher in ISCs and Paneth cells when treated with EIPA compared to control at day 1. Why did the authors use a pharmacological inhibitor rather than a siRNA or a dox-inducible CRISPR to inhibit NHE1 activity to look at the pH dynamics?

- Why is neck region not included in the EIPA-treated data in Figure 1C?

- Figure 2C, D: Here, the authors used a dox-inducible CRISPR to inhibit the activity of NHE1. However, the authors don't show/control the pH gradient in this genetic context. Also, did the authors confirm that the deletion worked by looking at the protein level?

- Figure 2G: The authors conclude there are no changes in the number of Paneth and Cd44+ cells. This is contrary to the conclusions made about the same cells in the same experimental conditions in Figure 4A-C and G. Ultimately, one of the main claims of the paper is that Paneth cells are affected by PH gradients. This is confusing. Is the total number of Cd44 cells not changed but the CD44+/LGR5+ cells changed? The data in Figure 2G should be quantified.

- Figure 4G-D: The authors report that NHE1 treatment leads to decreased proportion of LGR5+ cells but no change in proliferation as per EDU staining. However, they provide no possible explanation on this. Given that budding of organoids is affected, what is the meaning of this? Is the growth of organoids impacted by changes in PH? Have the authors tried other measures of proliferation? For example, Ki64? It is well recognised that the loss of Lg5 cells is not sufficient to impair intestinal proliferation. However, a better characterization of organoid growth or proliferation, beyond crypt formation, is important to better understand the functional importance of the PH gradient in the intestine.

- I find the single cell RNA sequencing data problematic/difficult to interpret. RT-qPCR and immunostaining do not appear to coincide/correlate with this data. For example: based on this information, the authors conclude that increase absorptive cell lineage and decrease

secretory lineages are produced following inhibition of PH gradient. However, cellular markers for the former cells are not changes in the RNAseq. How is it that these cells are classified/identified in the RNAseq data if it is not by their well-known molecular markers? How can one conclude changes in their differentiation if key identifiers are not changed? Specific examples below:

- Figures 5-6: In the text, authors claim “Consistent with these data, we observed increased Alpi1 transcript levels in extracts from organoids with inhibited NHE1 activity compared with controls as determined by qRT-PCR” p8/9 without citing any data. Do they talk about the data shown in Figure 6D? If yes, the gene expression fold change shown does not seem to be significantly different. Is it the case?

- Then, the authors claim that there is “no change observed in the average expression level of enterocytes signature genes Alpi1, Aldolase B (Aldob), Apoa1, and Apoa4 in the enterocyte clusters with inhibited NHE1 activity”. They should cite the figure 5D here and not 6D. I also find this type of data representation (Figure 5D) very difficult to interpret. Dot plots, including statistics would be better. For example, Apoa4 does not seem to be the same in all conditions.

- Figure 5G-I: Authors performed lineage tracing and counted the number of ALDO B+, RFP+ cells. They claim there is a slight increase of RFP+ ALDO B+ enterocytes in EIPA treated organoids at day 2 and 3. This quantification has been done by counting the number of cells RFP+, ALDO B+ cells/organoids. Authors should quantify the ratio cells RFP+, ALDO B+ cells/total number of RFP+ positive cells. Indeed, by looking at the picture in Figure 5G, it seems that there are more RFP positive cells when treated with EIPA. Based on these observations/comments, the conclusion made “there is a robust production of absorptive progenitors in organoids lacking NHE1 activity” is not well supported by the presented data. The ALDO B staining shows a lot of background making difficult to know when a cell is positive or not. Is there any other marker that could be use? This experiment is key to conclude if it is or not the case that lack of NHE1 activity promotes differentiation towards the absorptive lineage.

- Figure 6: The authors then looked at the secretory lineage while inhibiting NHE1. Is it possible to see the present here the same type of figures show in 5A-B? Are the changes shown in Figure 6B significant? Compared to Figure 5C for looking at enterocytes, the changes here seem milder. For both 6B and 5C, authors should show if changes are significant or no. This could be done by showing the result for each replicate with a dots plot representation.

- Figure 6C: after induction with doxycycline, it seems that there is an increase of Hes1 and Atoh1. Same for hes1 when adding EIPA. The conclusion made in the text seems in contradiction with this.

- Figure 6D: Author should mention if the changes are significant or not. Those experiments are key to conclude if there is a favour differentiation into absorptive or secretory lineage upon NHE1 inhibition. As it has been done for Paneth cells. Staining with markers of other ssecretory lineages (e.g. Chromogranin A for EE cells and Alcian Blue for Goblet Cells) should be done and quantified. A more detail analysis of the different secretory cell subpopulations from the RNAseq should be provided.

- Figure 6G-H: The authors claim that there is no difference of the number tdTomato+ cells at day 1. There is a mistake: it's at day 2.

- Figure 8 and Figure S7B: Authors show a decrease of EPHB2, which is supposed to be active by Wnt signalling, when NHE1 is inhibited. However, when they looked at Wnt targets

Axin2 and Ascl2, they saw a slight increase of their expression when NHE1 is inactivated. How can they explain this?

Minor points:

- Figure 1D: the pH in the cells at the base of the crypt after 49h in Figure 1D show quite a high pH compared to the pictures shown in Figure 1B: how do the author explain this difference?
- Video S2A not cited in the text.
- Figure S3D cited in the text, p7 but no figure S3D.
- Figure 7D: did the Authors look at day3?
- Figure 7H: The Author show the ratio of Lyz+/CD44+ cluster length. Why not the number or ration of cells per crypt?

Reviewer #3 (Remarks to the Author):

The work by Liu et al addresses the role of an intracellular pH gradient existing along the crypt-axis in the establishment of cell fate decisions in the gut. Although the study addresses a very interesting concept, there are numerous details and inconsistencies that needs to be addressed.

- The study refers to the previous work (Nikolovska et al in Acta Physiologica) which described an identical pH gradient in the colon. These authors describe an important role of NHE2 in establishing this gradient, rather than NHE1. NHE2 is also expressed in the small intestine, and the study would benefit from exploring the expression level of these transporters better in the organoid system and assessing their relative contribution to the gradient. The authors state EIPA is a selective pharmacological inhibitor of NHE1, but to my knowledge it is not. Moreover, the authors should discuss the seemingly opposing results that Nikolovska et al obtained in terms of the effect of intracellular pH on secretory vs absorptive cell ratio. If this is because of SI vs colon differences, this would be very interesting to confirm.
- Single cell RNA seq analyses would benefit from better visualization of data. For example, relative abundances of cell populations in the different treatments should be added in a single figure panel, as well as differential gene expression analysis within cell populations in the different treatments. From Figure 3 I understand all cells are present but cannot derive any meaningful information otherwise. It is not easy to have an overview when every cell lineage is addressed in separate Figures.
- Baseline Lgr5 levels are lower in non-Dox treated organoids harboring inducible NHE1 KO – as low as EIPA-treated ones. Could the authors discuss this, I understand there is no leakiness of guide expression based on the clean genotyping of WT organoids.
- The pH gradient is not validated in vivo/ex vivo on freshly isolated crypts. This would strengthen the story.
- Authors make seemingly contradictory statements on Figure 5: first “ While this difference is not statistically significant, the trend is consistent with the predictions made by scRNA-seq analysis indicating that inhibiting NHE1 activity does not impair differentiation to enterocytes. T” and next “ Collectively, our data indicate that inhibiting NHE1 activity promotes differentiation toward the absorptive lineage.” I believe the authors cannot conclude now that

NHE1 inhibition promotes differentiation towards the absorptive lineage based on the lineage tracing (cannot explain doubling of the population); selective killing of secretory cells would also cause a relative increase in the single cell rna sequencing data of enterocytes.

- The authors conclude for Figure 5D that there is no change in enterocyte marker expression upon loss of NHE1 activity. It does seem from these plots that lower APOA1/4 expressing cells reduce and highest APOA1/4 expressing cells increase with EIPA/dox. Again for these aspects unbiased differential gene expression analysis within populations would add.

- The authors write " Therefore, the loss of crypt budding with inhibiting NHE1 activity is likely caused by the reduction in Lgr5+ ISCs." To sum up Figure 4. Is it more likely that the impaired WNT secretion is causing defective budding and reduction in ISCs separately? I am not certain how fewer ISCs would completely block budding.

- The authors state : " Likewise, we also observed decreased expression of these genes in both the Paneth-Goblet progenitor and the Goblet precursor populations (Figure 6C), supporting our observation that the secretory cell population but not the ATOH1+ secretory progenitor cell population is reduced (Figure 6B)" I agree based on Figure 6 that ATOH1 targets DLL1 and SPDEF decrease with NHE1 loss. First, it would be useful to add more ATOH1 targets to the analysis. Second, it would be useful to separately show Paneth cells and Goblet cells in these analyses.

- The authors would have to exclude that secretory cells do not selectively get killed upon EIPA treatment. Do organoids become refractory to Notch inhibition (ATOH1 activation) when co-treated with EIPA?

- I do not understand the lineage tracing experiment in Figure 5, using the ATOH1 tracer. The authors state the Tomato+ cells are equal at d1, but only increase in non-treated organoids. The expectation would be – if I understand correctly – that numbers of traced cells should not be different between treatment and non-treatment if pH is acting downstream of ATOH1. ATOH1+ cells are also not expected to proliferate extensively, or do the authors propose that it is rather expansion by proliferation of secretory progenitors that is affected (should be shown). An alternative experiment could be to perform tracing and stain all secretory cell markers within ATOH1-traced cells – as done for PCs. These should be reduced, if maturation downstream of ATOH1 is affected.

- The authors state that canonical WNT in ISCs might be increased in EIPA treated organoids, and show Axin2 and ASCL2. However, LGR5 expression in individual stem cells is reduced (Figure 4), another WNT target gene. How solid is this observation for a broad panel of Wnt target genes?

Minor:

- One typo: Figure 6D is referenced when 5D should be.

- The authors should test that Cas9-GFP induction has no effect on epithelial differentiation dynamics.

- Figure 5C: axis labeling is not intuitive

- Are enteroendocrine cells affected by NHE1 inhibition? The numbers do not change based on Figure 5B; but there is a reduction in CHGA in Figure 5D.

- Figure S5 would benefit from Lysozyme stainings to convincingly show that Lgr5 cells do not form new Paneth cells upon treatment.

- In Figure 7G I have the impression that organoid growth in EIPA treatment is actually reduced compared to non-EIPA. Could the authors depict Paneth cell numbers as a total of organoid cell numbers?

Reviewer #1 (Remarks to the Author):

Major points:

1. It was recently shown that intestinal stem cells display rather dynamic metabolism, which can be strongly dependent on the availability of the metabolic substrates in the medium - glucose, pyruvate and perhaps others (Okkelman et al, Redox Biology, 2020, 30, 101420) and thus it is important to see if e.g. glucose present in the medium would affect the observed pHi gradients. This could happen if e.g. stem cells activate their OxPhos. Therefore I would be interested to see if the observed crypt-villi gradient of pHi is (not) dependent on cell metabolism and availability of the glucose in the medium. I see that authors used buffer supplemented with 10 mM Glc (with bicarbonate, 115 mM NaCl etc.) in the experiments but it also does not contain pyruvate, glutamine amino acids etc. How long the incubations were performed? Do you think that pHi in the intestinal crypt cells would change with duration of incubation in the experimental buffer? Please discuss and provide experimental evidence, if possible.

We thank the Reviewer for their comments. We agree that it would be interesting to learn more about the relationship between the metabolic profile of each crypt cell type and their pHi, but we believe that this is a separate question beyond the scope of the current study. Our current focus is to demonstrate that there is a pHi gradient in intestinal organoids and that this has a role in lineage specification. The factors establishing the pHi gradient are clearly important to resolve in a future study. Notably, the microscopy was performed immediately after moving the organoids into a bicarbonate-containing buffer (including 10 mM glucose) from the growth medium to reduce the intervention from buffer exchange. In general, organoids were incubated in the bicarbonate buffer for less than 1 h during the entire imaging acquisition. We repeatedly observed a similar pHi gradient in bicarbonate buffer (including 10 mM glucose) (Fig. 1b) as well as in the complete growth medium (including pyruvate, glutamine, and 18 mM glucose) (Fig. 1d), suggesting that our approach (buffer exchange and incubation duration) was appropriate for quantifying pHi and reliably reflected pHi dynamics in the growth condition.

2. Can the inhibition of NHE1 be not 100% specific with EIPA? Please discuss or perhaps add experimental evidence e.g. with dimethyl amiloride or other inhibitor, e.g. from the article: <https://bpspubs.onlinelibrary.wiley.com/doi/full/10.1111/bph.14429>

At the 5 μ M concentration we used, EIPA is a highly selective inhibitor of the NHE1 isoform (PMID: 11807182). We include data below for Reviewer 1 showing complete loss of NHE1 activity in human colorectal HCT116 cells treated with 5 μ M EIPA. As standardly used, NHE1 activity is determined by measuring the rate of pHi recovery (dpHi/dt) from an NH₄Cl-induced acid load. In brief, cells loaded with the pHi-sensitive dye BCECF are incubated with 40 mM NH₄Cl for 10 min, and then rapidly washed with a Hepes buffer (nominally HCO₃⁻-free to block H⁺ extrusion from HCO₃ transporters). The time-dependent increase in pHi is an index of H⁺ extrusion by NHE1. As

indicated in the figure, control cells have a rapid and robust increase in pHi. In contrast, cells treated with EIPA or engineered with CRISPR/Cas9 to silence NHE1 expression (NHE1 KO) have no pHi recovery, indicating complete loss of NHE1 activity. This experimental procedure cannot be performed with organoids because it requires a rapid exchange of buffers, which cannot be achieved within seconds in multilayers of cells embedded in Matrigel.

Minor points:

1. Page 12, "for the Paneth cell marker, LYZ" - add reference for lysozyme as a marker.

As requested, our revision includes the reference (Sato et al., 2011, PMID 21113151).

2. Page 21. "5 μ M EIPA for 1 to 3 days." - how this was optimised? Perhaps I am missing it somewhere?

10 μ M is a standard concentration of EIPA used to inhibit NHE1 activity in clonal cells. We initially tested 5 and 10 μ M EIPA on organoids and found that 5 μ M was sufficient to impair budding without affecting epithelial viability. In addition, as shown above, 5 μ M effectively blocks NHE1 activity in clonal intestinal cells, Therefore, we used 5 μ M EIPA throughout our study. The time points of 1 and 3 days were selected based on our observing the time it takes for WT organoids to develop crypts. In response to this comment, we added this information to the methods as follows *"To inhibit NHE1 activity, 5 and 10 μ M EIPA were tested with WT organoids. The 5 μ M EIPA was selected based on its efficacy in blocking crypt budding and its effect on epithelium viability. As the crypt development took 3 days, day 1 and day 3 were selected for capturing crypt budding. In brief, organoids embedded in Matrigel were cultured in ENR medium supplemented with DMSO vehicle (1:1,000) or 5 μ M EIPA for 3 days, and a microscopy-based measurement was performed on day 1 and day 3 organoids."*

3. Page 21. Below, description of membrane dye CellMask. Please provide used concentration.

We used CellMask at 1:1000, as per the manufacturer's instructions. As requested, we added this clarification in the Methods section.

4. Page 22. "segmented by the membrane dye." - which dye? CellMask?

Yes, that is correct. The text has been edited as *"Individual cells were segmented by CellMask™ dye"*.

5. Page 22. "Because of the mCherry/SEpHluorin intensity ratio in the cells in the organoid villus (above the..." - can you provide more specific values and perhaps S/N ratios?

In response to this comment, we modified the text as follows: “As the villus thickened and enlarged as a spheroid, background signals started to mask the actual fluorescence from mCherry and SEpHluorin. This led to an unreliable mCherry/SEpHluorin ratio and resulted in a higher-than-normal pHi value (>7.8) that was out of the linear range of SEpHluorin (6.7-7.7). Therefore, pHi in the villus cells was not reported.”

Reviewer #2 (Remarks to the Author):

Major points:

1. This work is entirely carried out in vitro, using 3D cultured organoids. While this is a very valuable reductionist model, the main claims of this study, that there is an intracellular PH gradient in the crypt axis of the intestine that is functionally important for ISC differentiation need to be demonstrated in vivo to really confirm that the observations of this study are not influenced by culture conditions. For example, PH could be assessed in freshly dissected or fixed intestinal tissue samples.

As requested, we include new data (left; Fig. 1e,f; Supplementary Fig.1d) confirming a pHi gradient in freshly isolated crypts prepared from the small intestine of 10-week old adult WT mice. We added the following description in the main text “To determine whether a similar pHi gradient exists within the intestinal epithelium, we isolated whole crypts from freshly dissected

intestines and assayed for pHi using a ratiometric pH-sensitive dye. Indeed, we observed a clear pHi gradient in these freshly isolated crypts, with increasing values extending from the base of the crypt to the top (Fig. 1e,f; Supplementary Fig. 1d)..” To obtain these data, we loaded freshly Isolated crypts with a fluorescent pH sensitive dye, BCECF, followed by imaging and pH calibration. Ratiometric measurements indicated a pHi gradient with mean pHi ranging from 6.9 - 7.2 (bottom – top) along the axis of the isolated crypts. These estimates of pHi are lower than the estimates we obtained from organoids, likely because of the differences in the sample preparation of isolated crypts and the use of pH sensitive dye rather than mCherry::pHluorin. Nonetheless, they provide strong new support for a pHi gradient in the intestinal crypt.

2. The functional role of NHE1 could be addressed by conditional gene knockout using Villin-Cre. I believe this will be the most robust way to confirm the results and to look at the impact of PH in intestinal cell differentiation.

A mouse line with a conditional allele of NHE1 is not available. However, one of the advantages of working with organoids is that it allowed us to generate a CRISPR-based conditional knockout using lentiviral infection. This is functionally equivalent to using organoids from a mouse with a conditional knockout allele, and our results with the CRISPR-based conditional knockout provided strong support for our conclusions.

3. Figures 1B-1C: The conclusions made in the texts are not supported by the pictures shown in the figure. The authors conclude that there is a significantly lower pH in ISCs and adjacent non-Paneth cells when treated with EIPA compared to controls at day 3: this is difficult to see on the pictures shown in Figure 1B. It seems to be the contrary on the picture. Also the PH in Paneth cells at day three with and without EIPA seems to be non-significantly different as per the statistics shown in Figure 1C. It is also indicated that the pH remains unchanged in ISCs between controls and EIPA-treated organoids at day 1. On the picture shown in Figure 1B, it seems to be the contrary: the pH seems to be higher in ISCs and Paneth cells when treated with EIPA compared to control at day 1.

We thank the Reviewer for their comments. The pHi in a ratiometric image is indicated by the color scale, not the brightness, but the differences in brightness of the signal (due to differential expression of the pH sensor determined by pH-insensitive mCherry fluorescence) could impact visualization of the color scale. For Fig. 1b day 3 -EIPA, because its original dynamic range was wider than can be displayed in the figure, the overall brightness of this image was adjusted to avoid oversaturating signal in the neck region in day 3 -EIPA. However, this also resulted in a reduced signal, which as indicated by the comment is confusing. To address the Reviewer's comment, we added separate panels at the bottom of Fig. 1b that show enlarged views of each cell type of interest, set to comparable brightness levels. In a similar manner, we made changes on the brightness for day1 images (-/+ EIPA) in Fig. 1b, so their visual appearance represented our quantification in Fig. 1c. The Reviewer is correct that pHi in day 3 Paneth cells was not significantly different between control and EIPA, which we did not describe in the main text. In response to this comment, we added following description "*The pHi of EIPA-treated mature Paneth cells was lower at day 1 but unaffected at day 3 (Fig. 1b,c)*".

4. Why did the authors use a pharmacological inhibitor rather than a siRNA or a dox-inducible CRISPR to inhibit NHE1 activity to look at the pH dynamics?

We agree that measuring pHi after genetically inhibiting NHE1 could be an approach in addition to our measuring pHi after pharmacological inhibition of NHE1. However, generating a CRISPR knockout is not currently feasible in a pHluorin expressing background which would interfere with Cas9-GFP that we use to generate dox-inducible silencing.

5. Why is neck region not included in the EIPA-treated data in Figure 1C?

We were unable to measure the pHi in the neck region of EIPA-treated organoids because the crypts do not form stably enough to have a clear neck region under these conditions. In response

to this comment, we now state “Because the crypts in EIPA-treated organoids did not form buds that were stable enough to have a clear neck region, the pH of crypt neck in those organoids was not measured.”.

6. Figure 2C, D: Here, the authors used a dox-inducible CRISPR to inhibit the activity of NHE1. However, the authors don't show/control the pH gradient in this genetic context. Also, did the authors confirm that the deletion worked by looking at the protein level?

As indicated in our manuscript, we confirmed silencing by DNA sequencing. The abundance of endogenous NHE1 is generally too low to detect. Indeed, our single cell RNA-seq did not detect NHE1 in control organoids, and protein abundance cannot be detected by immunoblotting or immunofluorescence with the available NHE1 antibodies.

7. Figure 2G: The authors conclude there are no changes in the number of Paneth and Cd44+ cells. This is contrary to the conclusions made about the same cells in the same experimental conditions in Figure 4A-C and G. Ultimately, one of the main claims of the paper is that Paneth cells are affected by PH gradients. This is confusing. Is the total number of Cd44 cells not changed but the CD44+/LGR5+ cells changed? The data in Figure 2G should be quantified.

Please see Fig. 7h for a quantification of the number of Paneth cells as a function of the length of the CD44+ expression domain, which is shown in Fig. 2g. The purpose of showing the images in Fig. 2g is not to support a claim about the number of Paneth cells or the CD44+ cells. Instead, the purpose of this figure is to show that we detected an unbudded crypt region with loss of NHE1 activity, as evidenced by our observation that Paneth cells were still present as a cluster in the CD44+ cluster. We edited the text to “We were able to detect CD44+ cell clusters in organoids that lack NHE1 activity (Fig. 2g)” to make our point clearer. In organoids, CD44+ cell number (crypt region size) usually varied dramatically between individuals, so we used LGR5+ cell number/CD44+ cell number to normalize this variation for quantification.

8. Figure 4G-D: The authors report that NHE1 treatment leads to decreased proportion of LGR5+ cells but no change in proliferation as per EDU staining. However, they provide no possible explanation on this. Given that budding of organoids is affected, what is the meaning of this? Is the growth of organoids impacted by changes in PH? Have the authors tried other measures of proliferation? For example, Ki64? It is well recognised that the loss of Lgr5 cells is not sufficient to impair intestinal proliferation. However, a better characterization of organoid growth or proliferation, beyond crypt formation, is important to better understand the functional importance of the PH gradient in the intestine.

As requested, we include new data with Ki67 staining, with results that support our findings with Edu staining. These new data are included in Supplementary Fig. 4c.

We agree that “It is well recognized that the loss of *Lgr5*+ cells is not sufficient to impair intestinal proliferation”. Thus, we would not expect that the decrease in *Lgr5*+ cells would be associated with reduced proliferation. In addition, because the majority of proliferative cells in organoids are not *Lgr5*+ ISCs, it is not surprising that a decrease in the number of *Lgr5*+ cells would not have a significant impact on the overall rate of proliferation in crypts.

For the budding phenotype, the loss of *Lgr5*⁺ cells is confirmed to lead to loss of budding in organoids (PMID 33503423). Therefore, although the loss of *Lgr5*⁺ ISC did not impact overall proliferation in crypts, it can impair crypt budding. Our data combined with previous findings suggest that budding is a process distinct from proliferation.

In a response to this comment, we revised the text to *“The loss of Lgr5⁺ ISCs is known to cause crypt loss in organoids (Tan et al., 2021), but is not sufficient to impair proliferation in the intestinal epithelium (Tetteh et al., 2016 PMID 26831517; Tian et al., 2011 PMID 21927002; van Es et al., 2018 PMID 23000963). Consistent with this, we observed no difference in the frequency of EdU⁺ and Ki67⁺ cells in organoids maintained without or with EIPA (Fig. 4e,h; Supplementary Fig. 4c). Together, these findings indicate that inhibiting NHE1 activity causes a decrease in Lgr5⁺ ISCs without impacting proliferation of crypt cells.”*

9. I find the single cell RNA sequencing data problematic/difficult to interpret. RT-qPCR and immunostaining do not appear to coincide/correlate with this data. For example: based on this information, the authors conclude that increase absorptive cell lineage and decrease secretory lineages are produced following inhibition of PH gradient. However, cellular markers for the former cells are not changes in the RNAseq. How is it that these cells are classified/identified in the RNAseq data if it is not by their well-known molecular markers? How can one conclude changes in their differentiation if key identifiers are not changed? Specific examples below:

qRT-PCR and scRNA-seq are complementary, partially overlapping methods for testing whether an experimental perturbation affects cellular differentiation, and we agree that it can be more difficult to interpret the data when the results from the two approaches are not in agreement. In response to this comment, we have softened our conclusion about the effect that loss of NHE1 function has on specification toward the absorptive lineage because we could not confirm our scRNA-seq observations by qRT-PCR in this case.

In scRNA-seq analysis, cells are clustered based on a broad comparison of all genes detected in each cell and analysis of the mean expression of cell type specific genes is used to determine which cell type each cluster corresponds to. Clustering cells based on all genes detected in each cell is a robust method of cell type identification that does not require every cell in the cluster to express a particular cell type marker, and a change in cluster size (expressed as a proportion of the total cell population) between control and experimental conditions is a good indicator of an effect on the population size of the cell type that is represented by that cluster.

10. Figures 5-6: In the text, authors claim *“Consistent with these data, we observed increased Alpi1 transcript levels in extracts from organoids with inhibited NHE1 activity compared with controls as determined by qRT-PCR”* p8/9 without citing any data. Do they talk about the data shown in Figure 6D? If yes, the gene expression fold change shown does not seem to be significantly different. Is it the case?

In a response to revise the manuscript, we changed the order of presenting secretory and absorptive lineages. Original Figure 5 is now Fig.6, and the old Figure 6 has become Fig.5.

Yes, we were referring to Fig. 5d (original Figure 6D). We apologize for the confusion and our revision cites the data. A statistical analysis of the data in Fig. 6d indicates that the mean expression values of *Alpi1* in the control versus NHE1-inhibited condition, as assessed by qRT-PCR of bulk tissue, is not significant. Although our scRNA-seq analysis, via analysis on multiple markers, predicted an increase of enterocyte differentiation with loss of NHE1 activity (Fig. 6b,c: increase of cell numbers classified as enterocyte. Fig.6d: increase of enterocyte markers), our current validation approach using lineage tracing and qRT-PCR of a single enterocyte marker did not confirm this increase. Therefore, we have softened our statement by modifying the texts accordingly as per below:

For scRNA-seq:

“In addition, we observed a small but significant increase in the mean expression levels of Alpi1, Aldolase B (Aldob), and Apoa1 within the enterocyte clusters of the EIPA+ dataset compared to control (Fig. 6d). Similarly, we detected a significant increase of Alpi1, Aldob, Apoa1, and Apoa4 transcripts in the enterocyte clusters with NHE1 silencing (Fig.6d).”

For lineage tracing and qRT-PCR:

“However, we were not able to detect a significant difference in Alpi1 transcript levels by qRT-PCR using extracts from entire organoids with or without EIPA treatment (Fig.5d). As an additional test for an effect of EIPA treatment on specification of the absorptive cell specification, we performed lineage tracing with Lgr5CreER;Rosa26RFP organoids (Castillo-Azofeifa et al., 2019), which traces the differentiation of progeny from Lgr5+ ISCs into all cell types of the lineage (Supplementary Fig.5a), including enterocytes (Fig. 6g), secretory progenitors (Fig. 5i), and Paneth cells (Fig.7a). We observed no difference in the number of enterocytes, marked by Aldob expression (Lukonin et al., 2020; Serra et al., 2019) with in RFP-labeled clones in EIPA-treated organoids compared with controls (Fig. 6g-i; Supplementary Fig. 5c). Taken together, these results indicate that inhibiting NHE1 activity does not impair absorptive cell differentiation and might induce a slight bias in differentiation toward this lineage.”

11. Then, the authors claim that there is “no change observed in the average expression level of enterocytes signature genes *Alpi1*, Aldolase B (*Aldob*), *Apoa1*, and *Apoa4* in the enterocyte clusters with inhibited NHE1 activity”. They should cite the figure 5D here and not 6D. I also find this type of data representation (Figure 5D) very difficult to interpret. Dot plots, including statistics would be better. For example, *Apoa4* does not seem to be the same in all conditions.

In a response to revise the manuscript, we changed the order of presenting secretory and absorptive lineages. Original Figure 5 is now Fig.6, and the old Figure 6 has become Fig.5.

We thank the Reviewer for these comments. We corrected the citation, and please refer to the response above for the text change.

As requested, we removed the original Ridgeplots in Fig. 6d (original Figure 5D). We realized that the Dot plot, however, was misleading in showing cells with certain marker expression such as *Aldob* (see example on the left). Instead, we provided violin plots and added statistical analysis (Fig. 6d). Indeed, the scRNA-seq data showed an increase in expression of enterocyte markers (such as *Alpi1*, *Aldob*, *Apoa1*) with loss of NHE1 activity. We also edited the text accordingly, and please see our response to the previous comment above.

12. Figure 5G-I: Authors performed lineage tracing and counted the number of ALDO B+, RFP+ cells. They claim there is a slight increase of RFP+ ALDO B+ enterocytes in EIPA treated organoids at day 2 and 3. This quantification has been done by counting the number of cells RFP+, ALDO B+ cells/organoids. Authors should quantify the ratio cells RFP+, ALDO B+ cells/total number of RFP+ positive cells. Indeed, by looking at the picture in Figure 5G, it seems that there are more RFP positive cells when treated with EIPA. Based on these observations/comments, the conclusion made “there is a robust production of absorptive progenitors in organoids lacking NHE1 activity” is not well supported by the presented data. The ALDO B staining shows a lot of background making difficult to know when a cell is positive or not. Is there any other marker that could be use? This experiment is key to conclude if it is or not the case that lack of NHE1 activity promotes differentiation towards the absorptive lineage.

In a response to revise the manuscript, we changed the order of presenting secretory and absorptive lineages. Original Figure 5 is now Fig.6, and the old Figure 6 has become Fig.5.

As requested, the enterocyte lineage tracing data have been re-analyzed and plotted as “# RFP+, ALDOB+ cells/ Total # RFP+ cells” (left; Fig. 6i). The new quantification, in line with our old analysis, shows no statistical significance in the enterocyte differentiation on both day 2 and day 3. Therefore, despite scRNA-seq predicting an increase of enterocyte differentiation, our validation via lineage tracing, which only examined one marker, was not capable of detecting this change. We think further investigation with more sophisticated approach is needed. Therefore, we modified our conclusion accordingly “Taken together, these results indicate that inhibiting NHE1 activity does not impair absorptive cell differentiation and might induce a slight bias in differentiation toward this lineage.”.

ALDOB is a robust and specific marker for staining enterocytes in organoids and has been used by several high-impact studies (PMID: 33029001; PMID: 31019299). The background of ALDOB staining comes from the dead cell shed in the lumen, and we could distinguish it from the real signal based on bright-field, Hoechst, and CD44 channels.

13. Figure 6: The authors then looked at the secretory lineage while inhibiting NHE1. Is it possible to see the present here the same type of figures show in 5A-B? Are the changes shown in Figure 6B significant? Compared to Figure 5C for looking at enterocytes, the changes here seem milder. For both 6B and 5C, authors should show if changes are significant or no. This could be done by showing the result for each replicate with a dots plot representation.

display replicates. Because quantifications of cell number changes in Fig. 5b and Fig. 6c were to generate a hypothesis for testing, also due to limited replicates of scRNA-seq (n=2), statistical analysis for these data are not provided.

14. Figure 6C: after induction with doxycycline, it seems that there is an increase of *Hes1* and *Atoh1*. Same for *hes1* when adding EIPA. The conclusion made in the text seems in contradiction with this.

In a response to revise the manuscript, we changed the order of presenting secretory and absorptive lineages. Original Figure 5 is now Fig.6, and the old Figure 6 has become Fig.5.

We added statistical analysis to this plot (Fig. 5c), and the expression of *Hes1* and *Atoh1* were not statistically different between controls and NHE1 inhibition, despite there was modest increase in *Atoh1* with Dox. We edited our text to reduce the confusion “*We found that expression of Hes1 and Atoh1 were similar in secretory progenitor cell populations from control and experimental datasets (Fig.5c), suggesting that inhibition of NHE1 activity does not impair the Notch pathway activity in these cells. In contrast, the expression of one or several ATOH1 key targets that are known to be involved in the secretory fate specifications, including Dll1, Spdef, Gfi1, and Neurog3 (Gerbe et al., 2011; Gregorieff et al., 2009; Lo et al., 2017; Noah et al., 2010; Shroyer et al., 2005) was decreased in the secretory progenitor, Paneth/Goblet progenitor, and Goblet precursor cell populations from EIPA-treated organoids (Fig.5c, Supplementary Fig.6a).*”

15. Figure 6D: Author should mention if the changes are significant or not. Those experiments are key to conclude if there is a favour differentiation into absorptive or secretory lineage upon NHE1 inhibition. As it has been done for Paneth cells. Staining with markers of other secretory lineages (e.g. Chromogranin A for EE cells and Alcian Blue for Goblet Cells) should be done and quantified. A more detail analysis of the different secretory cell subpopulations from the RNAseq should be provided.

In a response to revise the manuscript, we changed the order of presenting secretory and absorptive lineages. Original Figure 5 is now Fig.6, and the old Figure 6 has become Fig.5.

In a response to revise the manuscript, we changed the order of presenting secretory and absorptive lineages. Original Figure 5 is now Fig.6, and the old Figure 6 has become Fig.5.

As requested, we modified Fig. 5a (original Figure 6A) and added a split view that is similar to Fig. 6a,b (original Figure 5A,B) (also see example on the left). We also edited Fig. 5b and Fig. 6c to

As requested, our revision includes statistical analysis in Fig. 5d (original Figure 6D). Fig. 5d supports the hypothesis raised in Fig. 5b,c that secretory lineage differentiation, in general, was impaired with loss of NHE1 activity. Those data led us to validate this hypothesis via lineage tracing (Fig. 5j,k; Fig. 7b,c; Fig. 7e,f) and 3D immunostaining (Fig. 7h).

As requested our revision also includes new data on immunolabeling EEC with the marker *ChgA*. These new data, included in Supplementary Fig.6b, indicate fewer EEC with EIPA compared with controls and support our findings with lineage tracing.

16. Figure 6G-H: The authors claim that there is no difference of the number tdTomato+ cells at day 1. There is a mistake: it's at day 2.

In a response to revise the manuscript, we changed the order of presenting secretory and absorptive lineages. Original Figure 5 is now Fig.6, and the old Figure 6 has become Fig.5.

We thank the Reviewer for catching this. It was day 2 for organoid culture, but day 1 for lineage tracing. To avoid this confusion, we have edited it as follows “*We found no differences in the number of tdTomato+ cells in control and the EIPA-treated organoids on day 2 (Fig.5g)*”.

17. Figure 8 and Figure S7B: Authors show a decrease of EPHB2, which is supposed to be active by Wnt signalling, when NHE1 is inhibited. However, when they looked at Wnt targets *Axin2* and *Ascl2*, they saw a slight increase of their expression when NHE1 is inactivated. How can they explain this?

We agree that the role of NHE1 and pHi in Wnt signaling may not be straightforward as well as context dependent. We previously reported that there is an inverse relationship between NHE1 activity and pHi and Wnt pathway activity in naïve and differentiated mouse embryonic stem cells (PMID: 27821494). We also reported that increased pHi decreases Wnt pathway activity in mammalian and *Drosophila* epithelial cells in part by decreasing the stability of beta-catenin through increased affinity for beta-catenin binding the E3 ligase beta-TRCP1 (PMID: 30315137). And in unpublished findings we see that increased pHi attenuates binding of beta-catenin to Bcl9, the protein that mediates translocating beta-catenin from the cytosol to the nucleus. Additionally, Serafino et al., (PMID: 22387884) report that decreased NHE1 activity and pHi are linked to enhanced Wnt signaling in clonal colon adenocarcinoma cells, although a mechanism mediating this effect was not indicated, and Cruciat et al., (PMID: 20093472) report that intracellular acidification by inhibiting a vacuolar V-ATPase enhances Wnt signaling in *Xenopus* early development. Oginuma et al., (PMID: 32581357) show pHi regulation of non-enzymatic β -catenin acetylation downstream of WNT signaling in chick embryos mediated by an MCT monocarboxylate transporter. In contrast, Simons et al., (PMID: 19234454) reported that loss of a *Drosophila* NHE impairs Wnt signaling by attenuating membrane recruitment of dishevelled, which was selective for planar cell polarity and not seen for canonical Wnt signaling. Whether this is a pHi-dependent effect was not reported, but the study indicates a role for an NHE isoform in regulating some but not all aspects of Wnt signaling, which is a possibility for what we see with decreased EPHB2 but not *Axin2* or *Ascl2*. Our revision describes this possibility with regard to

these apparent discrepancies. Our revision includes new data showing decreased expression of a broad range of WNT targets (Supplementary Fig. 8c), including the catenin-independent WNT-responsive gene, *Klf5*. Altogether, the WNT signaling, in general, was reduced in ISCs despite some exceptions.

Minor points:

1. Figure 1D: the pH in the cells at the base of the crypt after 49h in Figure 1D show quite a high pH compared to the pictures shown in Figure 1B: how do the author explain this difference?

This difference was due to the imaging system. Fig. 1b used 40x immersive lens for viewing pH gradient with a high spatial resolution, while Figure 1d used 20x air lens for visualizing time-dependent dynamics. Because the optical properties are different between both lens and pH calibration was not feasible with timelapse imaging (Fig. 1d), the radiometric color scale would result a different appearance.

2. Video S2A not cited in the text.

We thank the Reviewer for catching this and have included reference to the video in the text “*In control organoids, crypts began budding by day 1, then lengthened and enlarged by day 3 as mature buds (Fig. 2a; Supplementary Video 2a)*”.

3. Figure S3D cited in the text, p7 but no figure S3D.

We thank the Reviewer for catching this error, it should be Supplementary Fig. 3c and has been corrected.

4. Figure 7D: did the Authors look at day3?

We did examine lineage tracing on day 3. As shown on the left, we tracked changes of Paneth cells in the same crypt across day 1, day 2, and day 3. Due to the challenge of time-lapse imaging, we could not reliably track cells in many crypts throughout day 3. In addition, we started to observe Paneth cell turnover between day 2 and day 3 (a,b), making this window day 2-day 3 not ideal for evaluating Paneth cell differentiation and

may complicate our interpretation. Furthermore, the Paneth cell number started to plateau (b), which was likely due to a combined effect of Paneth cell turnover and entry of lineage tracing late phase. Therefore, we chose window day 1-day 2 to assess Paneth cell differentiation for its reliability.

5. Figure 7H: The Author show the ratio of Lyz+/CD44+ cluster length. Why not the number or ration of cells per crypt?

We tried this method but realized it was not feasible to reliably count all CD44⁺ cells in 3D images due to the limitations of optics and Matrigel. There were too many CD44⁺ cells (except for the Paneth cells and ISCs which are restricted in small clusters), and they were small and touching each other. Combined with decreasing fluorescence signal in parts of the tissue that were further from the coverslip, it was not possible to obtain the necessary single-cell resolution for this experiment. For LYZ⁺ cells, however, it was relatively easy to distinguish as those are large and typically not adjacent to each other.

Reviewer #3 (Remarks to the Author):

1. The study refers to the previous work (Nikolovska et al in Acta Physiologica) which described an identical pH gradient in the colon. These authors describe an important role of NHE2 in establishing this gradient, rather than NHE1. NHE2 is also expressed in the small intestine, and the study would benefit from exploring the expression level of these transporters better in the organoid system and assessing their relative contribution to the gradient.

Our findings agree with those of Nikolovska et al., in showing a pH_i gradient – lower in crypts and higher in the villus region but differ in the NHE isoforms targeted to disrupt the gradient as well as effects on lineage specification. Whether there are differences in NHE1 regulated pH_i in the small intestine vs NHE2 regulated pH_i in the colon remains to be determined but we strongly believe this is outside the scope of our current study. Our data show that pharmacological and genetic inhibition of NHE1 attenuates the pH_i gradient and have robust and reproducible effects on intestinal stem cell differentiation and lineage specification. This does not rule out a role for NHE2 in pH_i dynamics in these organoids, but it does clearly show a role for NHE1. Nikolovska et al., report a role for NHE2 in pH_i dynamics in the colon but do not test ruling out a role for NHE1 and they “suggest that acid extruders other than NHE2 may play a more prominent role in pH_i regulation in the cryptal base and the cryptal mouth.” We show that loss of NHE1 activity decreases pH_i in both the crypt base and neck. The salient point is that there is a pH_i gradient in both tissues that regulates cell fate specification. However, the more substantial conflicting finding is that Nikolovska et al., report that loss of NHE2 decreases markers for enterocytes but increases Goblet cells and enteroendocrine cells, which we agree is distinct from what we found. The report from Nikolovska et al., however, does not include lineage tracing, which we include in our manuscript, and they indicate the caveat of possible compensatory changes with loss of NHE2 in their *in vivo* model. Compensatory changes with loss of NHE isoforms has been shown in multiple publications, which is why we chose to use inducible CRISPR/Cas9 silencing of NHE1.

2. The authors state EIPA is a selective pharmacological inhibitor of NHE1, but to my knowledge it is not. Moreover, the authors should discuss the seemingly opposing results that Nikolovska et al obtained in terms of the effect of intracellular pH on secretory vs absorptive cell ratio. If this is because of SI vs colon differences, this would be very interesting to confirm.

As indicated above in our response to comment 2 by Reviewer 1, EIPA at the 5 μ M concentration we used is a highly selective inhibitor of the NHE1 isoform and does not inhibit activity of NHE2 or other NHEs (PMID: 11807182). Also, as indicated in the figure included in our response to comment 2 by Reviewer 1 EIPA completely blocks NHE1 activity in human clonal colorectal HCT116 cells. However, in response to Reviewer 3's question, our revised discussion includes possible differences between SI and colon and a lack of consensus of our findings with those of Nikolovska et al. As per below *"Similar to our findings, an increasing pHi gradient is seen in murine colonic crypts (Amiri et al., 2021; Nikolovska et al., 2022). However, reducing pHi in colonic crypts is associated with decreased expression of absorptive fate genes and increased Goblet cell and Paneth cell markers (Nikolovska et al., 2022), which is in contrast to our findings in the small intestine. However, the small and large intestine are distinct in several ways, including the presence of protruded villi structure and Paneth cells, smaller crypt, more functional ISCs, and different transcriptional profiles (Azkanaz et al., 2022 PMID: 35831497), so the regulatory mechanism of ISC lineage responding to pHi dynamics in the small intestine may also be different. Comparing and contrasting the role of pHi in these two closely related organs will be an important topic for future study."*

3. Single cell RNA seq analyses would benefit from better visualization of data. For example, relative abundances of cell populations in the different treatments should be added in a single figure panel, as well as differential gene expression analysis within cell populations in the different treatments. From Figure 3 I understand all cells are present but cannot derive any meaningful information otherwise. It is not easy to have an overview when every cell lineage is addressed in separate Figures.

In a response to revise the manuscript, we changed the order of presenting secretory and absorptive lineages. Original Figure 5 is now Fig.6, and the old Figure 6 has become Fig.5.

We thank the Reviewer for the suggestion. We believe that starting our presentation of the scRNA-seq data with Fig. 3 to show the quality control analysis and the general approach we used for clustering is important. We also believe that separating analyses for ISC (Fig. 4), secretory lineage (Fig. 5), and absorptive lineage (Fig. 6) is clearer and for a more efficient and consistent data presentation. To address the Reviewer's concern and also as requested by Reviewer 2, we modified the scRNA-seq data visualization in Fig. 5a, Fig. 6d,e

4. Baseline Lgr5 levels are lower in non-Dox treated organoids harboring inducible NHE1 KO – as low as EIPA-treated ones. Could the authors discuss this, I understand there is no leakiness of guide expression based on the clean genotyping of WT organoids.

The comment is referring to the Fig. 4D. Because these are different genotypes, one is WT and the other is lentivirus-infected, so can't directly compare the baseline between them.

5. The pH gradient is not validated in vivo/ex vivo on freshly isolated crypts. This would strengthen the story.

As requested, we included new data (Fig. 1e,f; Supplementary Fig. 1d; see right; and also our response to comment 1 by Reviewer 2) using *ex vivo* on freshly isolated small intestinal crypts and observed a similar pH gradient.

6. Authors make seemingly contradictory statements on Figure 5: first “ While this difference is not statistically significant, the trend is consistent with the predictions made by scRNA-seq analysis indicating that inhibiting NHE1 activity does not impair differentiation to enterocytes. ” and next “ Collectively, our data indicate that inhibiting NHE1 activity promotes differentiation toward the absorptive lineage.” I believe the authors cannot conclude now that NHE1 inhibition promotes differentiation towards the absorptive lineage based on the lineage tracing (cannot explain doubling of the population); selective killing of secretory cells would also cause a relative increase in the single cell rna sequencing data of enterocytes.

In a response to revise the manuscript, we changed the order of presenting secretory and absorptive lineages. Original Figure 5 is now Fig.6, and the old Figure 6 has become Fig.5.

We thank the Reviewer for this comment. Staining with Caspase 3 did not reveal more cell death in the NHE1-inhibited condition compared to the control, but it may be due to the insensitivity of the assay. We agree that killing of secretory cells is a factor we cannot rule out. As per our above response to Reviewer 2’s comment--*Figures 5-6*, we modified our conclusion regarding absorptive lineage specification “*Taken together, these results indicate that inhibiting NHE1 activity does not impair absorptive cell differentiation and might induce a slight bias in differentiation toward this lineage.*” Additionally, we modified our statement for secretory lineage tracing (Fig.5f-h) to include the possibility of selective eliminating secretory cells “*although there was a net increase of tdTomato+ cells over time in control organoids, the number of tdTomato+ cells per crypt remained constant in organoids treated with EIPA (Fig. 5g,h; Supplementary Video 3a,b), suggesting that differentiation and clone expansion from Atoh1+ progenitor was attenuated.*”

7. The authors conclude for Figure 5D that there is no change in enterocyte marker expression upon loss of NHE1 activity. It does seem from these plots that lower APOA1/4 expressing cells reduce and highest APOA1/4 expressing cells increase with EIPA/dox. Again for these aspects unbiased differential gene expression analysis within populations would add.

In a response to revise the manuscript, we changed the order of presenting secretory and absorptive lineages. Original Figure 5 is now Fig.6, and the old Figure 6 has become Fig.5.

We thank the Reviewer for the suggestion. We replotted the data in Fig. 6d (original Figure 5D) using violin plots and performed statistics. We detected increased expression of all absorptive markers. We edited our text to *“In addition, we observed a small but significant increase in the mean expression levels of Alpi1, Aldolase B (Aldob), Apoa1 within the enterocyte clusters of the EIPA+ dataset compared to control (Fig.6d). Similarly, we detected a significant increase of Alpi1, Aldob, Apoa1, and Apoa4 transcripts in the enterocyte clusters with NHE1 silencing (Fig.6d).”*

8. The authors write *“ Therefore, the loss of crypt budding with inhibiting NHE1 activity is likely caused by the reduction in Lgr5+ ISCs.”* To sum up Figure 4. Is it more likely that the impaired WNT secretion is causing defective budding and reduction in ISCs separately? I am not certain how fewer ISCs would completely block budding.

Our study indicates that impaired Paneth cell differentiation and ISC maintenance in crypts leads to loss of budding. Previous studies (van Es et al., PMID:22393260; Tan et al., PMID: 33503423) showed decreasing the *Lgr5*⁺ ISC pool can block crypt formation in the organoid model. We agree with the Reviewer that other mechanisms might also contribute to this phenotype, as the mechanisms of crypt budding are still not fully understood. Accordingly, we modified the statement for Fig. 4 to read *“The loss of Lgr5+ ISCs is known to cause crypt loss in organoids (Tan et al., 2021), but is not sufficient to impair proliferation in the intestinal epithelium (Tetteh et al., 2016 PMID 26831517; Tian et al., 2011 PMID 21927002; van Es et al., 2018 PMID 23000963). Consistent with this, we observed no difference in the frequency of EdU+ cells and Ki67+ cells in organoids maintained without or with EIPA (Fig. 4e,h; Supplementary Fig. 4c). Together, these observations indicate that inhibition of NHE1 activity causes a decrease in Lgr5+ ISCs without impacting proliferation of crypt cells.”*

9. The authors state : *“ Likewise, we also observed decreased expression of these genes in both the Paneth-Goblet progenitor and the Goblet precursor populations (Figure 6C), supporting our observation that the secretory cell population but not the ATOH1+ secretory progenitor cell population is reduced (Figure 6B)”* I agree based on Figure 6 that ATOH1 targets DLL1 and SPDEF decrease with NHE1 loss. First, it would be useful to add more ATOH1 targets to the analysis. Second, it would be useful to separately show Paneth cells and Goblet cells in these analyses.

As requested, we added a separate panel showing additional ATOH1 transcriptional targets in Supplementary Fig. 6a (also shown on the top left). In general, the expression of secretory fate determination targets of ATOH1 (e.g. *Gfi1* and *Neurog3*) significantly reduced with the loss of NHE1 activity either by EIPA or CRISPR, while other targets exhibited a slight but not significant decrease. We agree with the reviewer that separating Paneth cells and Goblet cells in the scRNA-seq would be ideal. Unfortunately, we were not able to confidently distinguish these cell types from each other in our datasets. As demonstrated in the figure on the top right, many cells (red arrows) co-express Goblet and Paneth signature genes.

10. The authors would have to exclude that secretory cells do not selectively get killed upon EIPA treatment. Do organoids become refractory to Notch inhibition (ATOH1 activation) when co-treated with EIPA?

We thank the Reviewer for the suggestion. Our data suggest that the Notch signaling is not pH sensitive in this context, as indicated by our observation that Notch responsive gene expression (Fig. 6e: *Hes1* and *Olfm4* qRT-PCR) is not significantly different with or without inhibition of NHE1. In addition, in a pilot experiment, we tested the possibility that Notch signaling may account for the phenotypes we see in organoids with reduced NHE1 activity and compared the frequency of budding in organoids treated without or with EIPA alone or in combination with a potent Notch inhibitor, DAPT. We found that budding was inhibited to a similar extent in both conditions, suggesting that the lack of budding is not due to increased Notch signaling, so we did not pursue this direction further.

11. I do not understand the lineage tracing experiment in Figure 5, using the ATOH1 tracer. The authors state the Tomato+ cells are equal at d1, but only increase in non-treated organoids. The expectation would be – if I understand correctly – that numbers of traced cells should not be different between treatment and non-treatment if pH is acting downstream of ATOH1. ATOH1+ cells are also not expected to proliferate extensively, or do the authors propose that it is rather expansion by proliferation of secretory progenitors that is affected (should be shown). An

alternative experiment could be to perform tracing and stain all secretory cell markers within ATOH1-traced cells – as done for PCs. These should be reduced, if maturation downstream of ATOH1 is affected.

In a response to revise the manuscript, we changed the order of presenting secretory and absorptive lineages. Original Figure 5 is now Fig.6, and the old Figure 6 has become Fig.5.

We apologize for the confusion with our description of these data. Of note, the Reviewer was referring to the ATOH1 tracing experiment in Fig. 5f-h, which was in the original Figure 6F-H). We agree with the Reviewer that these data suggest that inhibiting NHE1 activity impairs expansion of the secretory progenitors downstream from *Atoh1* expression. We modified the text as follows: “*whereas there was a net increase of tdTomato⁺ cells over time in control organoids, the number of tdTomato⁺ cells per crypt remained constant in organoids treated with EIPA (Fig.5g,h; Supplementary Video 3a,b), implying differentiation and clone expansion from Atoh1⁺ progenitor was attenuated.*”

12. The authors state that canonical WNT in ISCs might be increased in EIPA treated organoids, and show *Axin2* and *ASCL2*. However, *LGR5* expression in individual stem cells is reduced (Figure 4), another WNT target gene. How solid is this observation for a broad panel of Wnt target genes?

As requested, we added a panel of additional WNT targets in Supplementary Fig. 8c (also shown on the right). In a summary, WNT targets, in general, decreased with the loss of NHE1 activity. As reported in this study, loss of NHE1 activity decreased Paneth differentiation to impair WNT pathway activation in the ISCs. However, loss of NHE1 activity and decreased pH_i have been reported elsewhere to enhance the canonical WNT pathway activity (White et al., 2018 PMID: 30315137; Serafino et al., 2012 PMID: 22387884; Cruciat et al., PMID: 20093472) via various cellular mechanisms. Therefore, we expected to see a mixed change in the expression of WNT pathway target genes in the ISCs. And indeed, we saw increased expression of the ISC signature genes *Ascl2* and unaltered expression of *Axin2*. We also observed down-regulations for the standard ISC marker *Lgr5* and a broad range of WNT targets, including the catenin-independent WNT responsive gene---- *Klf5*. In general, the data support our model in which the WNT signaling in ISCs was impaired with loss of NHE1 activity.

Minor:

1. One typo: Figure 6D is referenced when 5D should be.

We thank the Reviewer for catching the error and we corrected.

2. The authors should test that Cas9-GFP induction has no effect on epithelial differentiation dynamics.

The requested data is not standard for the field and not included in a vast number of publications that use intestinal organoids with inducible CRISPR/Cas9 silencing. Moreover, it took us nearly 6 months to generate a stable organoid line with inducible NHE1 CRISPR/Cas9 silencing and we strongly believe that adding the requested control does not warrant the delay in publishing our work.

3. Figure 5C: axis labeling is not intuitive

In a response to revise the manuscript, we changed the order of presenting secretory and absorptive lineages. Original Figure 5 is now Fig.6, and the old Figure 6 has become Fig.5.

As requested, we modified the axis labeling and the graphs (Fig.4c; Fig.5b; Fig.6c), and also edited the figure legends for clarity. An example is *"Fold change of enterocytes and Clu⁺ cells with NHE1 inhibition by EIPA compared to the control"*.

4. Are enteroendocrine cells affected by NHE1 inhibition? The numbers do not change based on Figure 5B; but there is a reduction in CHGA in Figure 5D.

In a response to revise the manuscript, we changed the order of presenting secretory and absorptive lineages. Original Figure 5 is now Fig.6, and the old Figure 6 has become Fig.5.

As requested, and see our response to a similar request by Reviewer 1, we include new data on immunolabeling enteroendocrine cells (EEC) with the marker *ChgA*. These new data, included in Supplementary Fig. 6b, indicate fewer EEC with EIPA compared with controls and support our findings with lineage tracing.

5. Figure S5 would benefit from Lysozyme stainings to convincingly show that Lgr5 cells do not form new Paneth cells upon treatment.

Please refer to Fig. 7a-c. We performed Paneth cell lineage tracing (labeled with Lysozyme) using *Lgr5^{CreER}*. We believe these are convincing data for Paneth cell differentiation.

6. In Figure 7G I have the impression that organoid growth in EIPA treatment is actually reduced compared to non-EIPA. Could the authors depict Paneth cell numbers as a total of organoid cell numbers?

Obtaining total cell number in organoids is not feasible due to the limitation of 3D imaging.

REVIEWER COMMENTS

Reviewer #1 (Remarks to the Author):

The manuscript of Nystul and co-workers reports the pH gradient across villi and crypt in the mouse intestinal organoid model. Authors have significantly revised the manuscript, added new data on the intestine, specificity of used inhibitor and, in my opinion, have addressed all the raised criticism. I thus recommend accepting this manuscript.

Reviewer #2 (Remarks to the Author):

The authors have addressed several of my concerns, which is commendable. However, there are still issues that have not been convincingly addressed.

1- New Figure 1e, f (Corresponding to Comment 1 from my review): I don't understand the information in this image. Not sure what structures are being depicted. Also, why did the authors not dissect whole intestines (crypt and villi) and assess PH gradient in such setting? This would reduce tissue manipulation involved in crypt isolation and provide a truly distinct setting to the one shown before, which is what I requested in my original comment where I asked for intestinal samples.

2- Most of the work remains to be essentially based on the use of the NHE chemical inhibitor and the authors are apparently unable to confirm mRNA or protein knockout in the NHE-CRISPR setting. I can to some extent understand the limitations. However, there are significant discrepancies between some of the phenomenology observed between the inhibitor treated and CRISPR edited organoids: effect on Lgr5 levels (Fig. 4d), various gene expression profiles (Figure 5, Supplementary Figure 8). This needs to be acknowledged at the very least.

3- In cases when the expression profile of a gene or panel of genes is used to drive a major conclusion, for instance, the role of the Ph gradient in stem cell lineage specification and Wnt signaling data from scRNA seq needs to be verified by RT-qPCR or immunostaining. This is not always the case

4- Related points 2 and 3 above: the data regarding the role of NHE in Wnt signaling activity is not robust to support the conclusions drawn. Is the data provided in Supplementary Figure 8C directly derived from the scRNAseq data? If so, this may have very little meaning. Acl2 appears significantly changed by scRNAseq data but not confirmed by RT-qPCR. Looking at nuclear B-catenin localization in the organoids and/or levels of some key/widely recognised targets of the pathway by immunostaining is a much more meaningful and needed to support the proposed working model.

Reviewer #3 (Remarks to the Author):

The rewriting / changing conclusions drawn from work have significantly improved the manuscript. I do still think inclusion of main gene expression changes per cell cluster would be valuable to include. I have no further comments otherwise.

REVIEWER COMMENTS

Reviewer #1 (Remarks to the Author):

The manuscript of Nystul and co-workers reports the pHi gradient across villi and crypt in the mouse intestinal organoid model. Authors have significantly revised the manuscript, added new data on the intestine, specificity of used inhibitor and, in my opinion, have addressed all the raised criticism. I thus recommend accepting this manuscript.

Reviewer #2 (Remarks to the Author):

The authors have addressed several of my concerns, which is commendable. However, there are still issues that have not been convincingly addressed.

1- New Figure 1e, f (Corresponding to Comment 1 from my review): I don't understand the information in this image. Not sure what structures are being depicted. Also, why did the authors not dissect whole intestines (crypt and villi) and assess PH gradient in such setting? This would reduce tissue manipulation involved in crypt isolation and provide a truly distinct setting to the one shown before, which is what I requested in my original comment where I asked for intestinal samples.

The representative image in Fig. 1e shows two isolated intestinal crypts (bottom, crypt base; top, crypt neck, as labeled in the figure) loaded with the pH-sensitive dye BCECF. We included these new data in response to the reviewers' request that we confirm our observations of pHi in organoids within intact tissue. We chose to perform the measurements of pHi in freshly isolated crypts rather than in dissected whole intestines because the crypt region is the part of the tissue that is the focus of our study and isolated crypts are much more amenable to deep, uniform BCECF dye penetration than whole tissue. Deep, even dye penetration is critical for obtaining the high-quality quantitative ratiometric imaging data that are needed to accurately quantify pHi in intact tissues. Notably, the measurements were taken within 60 minutes, and thus is a qualitatively distinct experimental setting than organoids, which grow into a tissue from a small number of cells over the course of several days. We strongly believe these new data effectively address the reviewers' request of confirming a pHi gradient in freshly isolated tissue.

2- Most of the work remains to be essentially based on the use of the NHE chemical inhibitor and the authors are apparently unable to confirm mRNA or protein knockout in the NHE-CRISPR setting. I can to some extent understand the limitations. However, there are significant discrepancies between some of the phenomenology observed between the inhibitor treated and CRISPR edited organoids: effect on Lgr5 levels (Fig. 4d), various gene expression profiles (Figure 5, Supplementary Figure 8). This needs to be acknowledged at the very least

We agree with this comment and have now added to the text the below sentence to acknowledge this limitation.

“The discrepancy is likely due to the distinct inhibitory mechanisms via EIPA and CRISPR, which are functionally complementary but not identical”.

3- In cases when the expression profile of a gene or panel of genes is used to drive a major conclusion, for instance, the role of the Ph gradient in stem cell lineage specification and Wnt signaling data from scRNA seq needs to be verified by RT-qPCR or immunostaining. This is not always the case.

We agree that it is important to verify scRNAseq results with additional experimental evidence. Indeed, the current manuscript includes descriptions of several independent approaches that support our conclusions from the scRNAseq experiments. For example, for our conclusion that disrupting the pHi gradient reduces WNT signaling we included (1) immunolabeling crypts for WNT targets LGR5 (Fig.4f,g) and EPHB2 (Fig.8a, Supplementary Fig.8b); (2) qRT-PCR for *Lgr5* in whole organoids; (3) a functional assay demonstrating addition of Rspodin1 and WNT3A rescues phenotypes caused by disrupting the pHi gradient (Fig.8b-e). Additionally, for our conclusion that disrupting the pHi gradient impairs lineage specification, we used comprehensive lineage tracing (Fig.5f-k; Fig.6g-l; Fig.7a-f; Supplementary Fig.6b), immunolabelling of secretory cell markers (Fig.7g-h; Supplementary Fig.7b), and qRT-PCR of signature secretory fate genes (Fig.5d,e). These experiments provide strong support using multidisciplinary approaches for the conclusions suggested by our scRNAseq data. However, in response to this comment, our new revision now includes the following sentence in the Discussion section to highlight the caveat raised by the reviewer. *“Given the complexity of WNT signaling and the high heterogeneity of intestinal epithelial cells, quantitative determination of pHi-WNT interplay along the crypt-villus axis will require a more sophisticated approach in future studies.”*

4- Related points 2 and 3 above: the data regarding the role of NHE in Wnt signaling activity is not robust to support the conclusions drawn. Is the data provided in Supplementary Figure 8C directly derived from the scRNAseq data? If so, this may have very little meaning. *Acl2* appears significantly changed by scRNAseq data but not confirmed by RT-qPCR. Looking at nuclear B-catenin localization in the organoids and/or levels of some key/widely recognises targets of the pathway by immunostaining is a much more meaningful and needed to support the proposed working model.

The original Supplementary Figure 8c is now Supplementary Figure 9c as a revision response to Reviewer #3's comment. Yes, the Supplementary Figure 9c is from the scRNA-seq data. We included this panel in response to a previous request from Reviewer #3. As described in our response to comment #2 above, we added a sentence acknowledging that NHE1 inhibition by EIPA and CRISPR are two complementary and distinct approaches that can cause discrepancies in the individual gene expression profiles. In addition, as we describe in our response to comment #3 above, we have, in fact, already included data from multiple independent approaches to test the effect of disrupting the pHi gradient on WNT pathway activity. The minor discrepancy between RT-qPCR (Supplementary Fig.9b) and the scRNA-seq data (Supplementary Fig.9a) is likely due to the complexity of WNT signaling and the sensitivity of the detection methods. Unfortunately, it is not possible to detect β -catenin in the nucleus of these cells by

immunofluorescence and, indeed, this is not standard in studies of Wnt signaling *in situ*. There remains a long-standing and still unresolved question in the field on distinct pools of cytoplasmic and nuclear β -catenin, in large part because scoring nuclear β -catenin remains an experimental challenge.

Reviewer #3 (Remarks to the Author):

The rewriting / changing conclusions drawn from work have significantly improved the manuscript. I do still think inclusion of main gene expression changes per cell cluster would be valuable to include. I have no further comments otherwise.

We thank Reviewer #3 for the suggestion. As requested, we now include the main changes in gene expression for ISC, secretory progenitor, Paneth/Goblet cells, EEC, tuft cell, and enterocyte clusters in the new Supplementary Fig.5.

REVIEWERS' COMMENTS

Reviewer #2 (Remarks to the Author):

The authors have provided textual responses to my remaining comments, including experimental limitations in the system. I have no further comments.

REVIEWERS' COMMENTS

Reviewer #2 (Remarks to the Author):

The authors have provided textual responses to my remaining comments, including experimental limitations in the system. I have no further comments.

We are pleased that all the Reviewers are now satisfied with the manuscript. We have made minor edits throughout to correct typos and improve clarity.